

# New approaches for capturing and estimating variation in complex animal color patterns from digital photographs: application to the Eastern Box Turtle (*Terrapene carolina*)

Erik Maki[1], Tilmann Glimm[2], Paramahansa Pramanik[3], Ylenia Chiari[1,4] and Maria Kiskowski[3]

[1] Department of Biology, George Mason University, Fairfax, VA, United States of America
[2] Department of Mathematics, Western Washington University, Bellingham, WA, United States of America
[3] Department of Mathematics and Statistics, University of South Alabama, Mobile, AL, United States of America
[4] School of Life Sciences, University of Nottingham, Nottingham, United Kingdom

Corresponding authors
Ylenia Chiari,
Ylenia.Chiari@nottingham.ac.uk
Maria Kiskowski,
abyrne@southalabama.edu

## ABSTRACT

Color pattern plays a crucial role in various aspects of an organism's biology, including camouflage, mating, and communication. Despite its significance, methods to quantify and study color pattern variation are often lacking, especially for complex patterns that defy simple categorization. In this study, we developed algorithms to capture and obtain data on 19 different pattern measurements from digital images of 55 individuals of the Eastern box turtle *Terrapene carolina* sampled in the field and in a museum. The Eastern box turtle is an ideal species to study variation of complex color patterns as this species is easily encountered in the field and in museum collections in Northeastern US, has a relatively easy to identify bright color pattern against a dark background, and has a rigid shell structure, which removes problems related to body distortion. The selected measurements capture the different aspects of the complexity of the color pattern, including the symmetry of the pattern on the turtles' scutes, a critical component in developmental and evolutionary studies. We estimated the variation of each of these 19 measurements across our samples. We determined how much of this variation was influenced by the sensitivity of the pattern capture algorithm due to non-standardized elements of the image acquisition, lighting conditions, and animal shape on pattern variation. To our knowledge, this is the first study to use a comprehensive set of pattern measurements to capture variation in a complex color pattern while also assessing the susceptibility of each of these measurements to noise introduced during data collection. Additionally, we carried out a citizen science approach to characterize the complexity of the color pattern based on human perception and determine which of the 19 pattern measurements best describe this complexity. The most variable measurements across individuals were blue and yellow contrast between the pattern and non-pattern coloration and the average size of objects. From our estimates of the measurement noise due to image acquisition and analysis, we found that the contrast differences reflected true pattern variations between individual turtles, whereas differences in the average size of objects were influenced by both individual turtle variation and measurement inconsistencies. We found that due to the complexity of the patterns, measurements

had lower variability if they did not depend on the algorithm defining a set of discrete objects. For example, total area had much less measurement variability than average object area. Our study provides a comprehensive workflow and tools to study variation in complex color patterns in organisms sampled under non-standardized conditions while also estimating the influence of noise due to biological and non-biological factors.

## INTRODUCTION

Animal color and color pattern are fundamental traits in ecology and evolution, frequently implicated in communication among individuals of the same species, sexual selection, anti-predator strategies, and thermoregulation. Consequently, they often undergo strong selective pressures (*Caro, 2005*; *De Solan et al., 2020*; *Gomez & Théry, 2007*; *Tibbetts & Dale, 2004*). Variation in color and pattern exists not only between species, life stages, and individuals, but also within different parts of the same animal, potentially in response to diverse selective forces (*Allen et al., 2020*; *Forsman et al., 2008*; *Glimm et al., 2021*). While extensive research has examined animal color patterns, particularly focusing on broad categories such as spots, bands, or stripes (*Endler, 1990*; *Hemingson, Cowman & Bellwood, 2024*; *Mason & Bowie, 2020*; *Kiskowski et al., 2019*; *Pérez-Rodríguez, Jovani & Stevens, 2017*; *Shamir et al., 2010*), the variation in size, shape, distribution, spatial organization, and other components of coloration of these patterns is still mostly overlooked (but see for example *Chan, Stevens & Todd, 2019*; *Glimm et al., 2021*; *Hastings et al., 2023*; *Miyazawa, Okamoto & Kondo, 2010*; *Stoddard, Kilner & Town, 2014*). However, neglecting these nuances could miss crucial aspects of pattern variation and its implication for ecological, developmental, and evolutionary processes. As such, more refined methodologies to capture these intricate differences are needed. Additionally, because the few approaches currently available to capture detailed measurements of the color pattern have not been applied to diverse organisms sampled under variable conditions, it is essential to assess how noise from systematic variation, such as differences in lighting and camera angles, could influence estimates of color pattern variation.

Describing and quantifying color pattern variation in detail poses a challenge, particularly in capturing and breaking down complex patterns into their constituent *pattern objects*— defined as distinct, separately identified parts of the pattern such as spots, stripes, or other discrete shapes—and *pattern measurements*, which are quantitative descriptors used to represent pattern complexity and variation. Identifying the most representative measurements to characterize the variation (*e.g.*, *Glimm et al., 2021*) is challenging as well. This is due to the fact that complex patterns contain objects and structures with elaborated edges and diverse or irregular shapes (*Stoddard & Osorio, 2019*) or are made up of objects that may be lightly connected (*e.g.*, Figs. 1 and 2). Numerous studies have examined

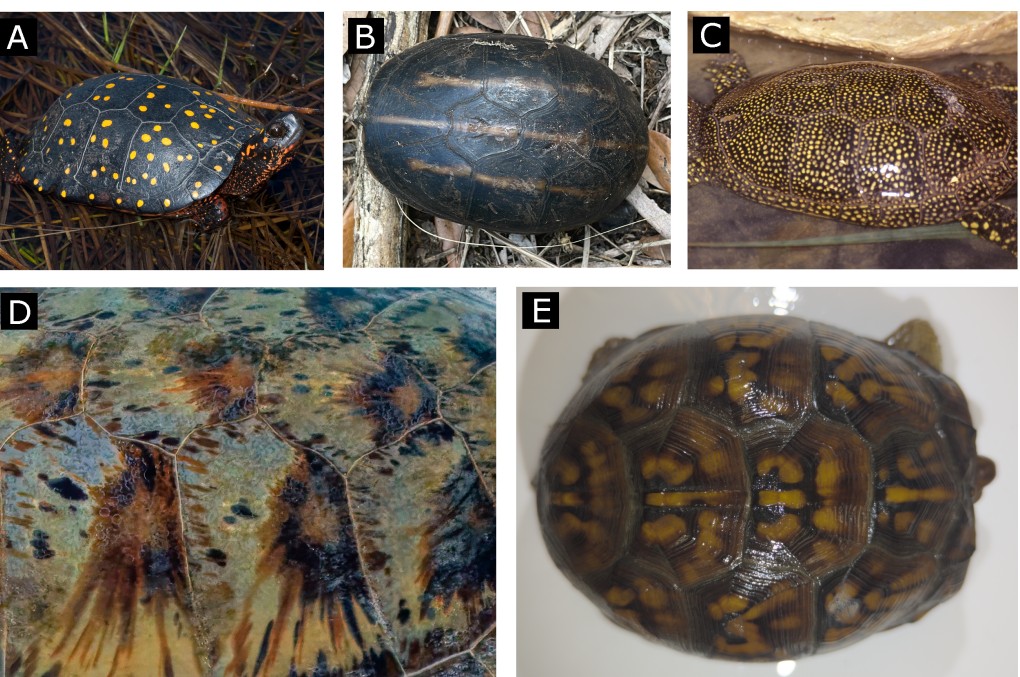

**Figure 1** **Examples of simple and complex color patterns (top row and bottom row, respectively).** These particular specimens of (A) *Clemmys guttata* (spotted turtle), (B) *Kinosternon baurii* (striped mud turtle) and (C) *Emys orbicularis* (European pond turtle) are characterized by patterns with uniform shapes (spots, stripes) and a relatively homogenous monocolor within each pattern object. In contrast, particular specimens of (D) *Chelonia mydas* (Green sea turtle) and (E) *Terrapene carolina* (Eastern box turtle) are characterized by a pattern with irregular shapes within each scute of the carapace and a broader range of blended, interconnected colors with no clear pattern organization on each scute. (Photo sources: (A) J. Vandermeulen CC BY-NC-ND (*Vandermeulen, 2009*), (B) sodancer CC BY-NC (*sodancer, 2025*), (C) B. Dupont CC BY-SA (*Dupont, 2018*), (D) C. Sharp CC BY-SA 4.0 (*Sharp, 2024*) (E) E. Maki, Smithsonian turtle with ID 22614).

variations in coloration metrics such as luminosity, contrast, reflectance, hue, saturation, brightness, and irradiance (*e.g.*, *Butler, Toomey & McGraw, 2011*; *Francini & Samia, 2015*; *Lorioux-Chevalier et al., 2023*; *Macedonia, Echternacht & Walguarnery, 2003*; *Stelbrink et al., 2019*). In contrast, aspects like pattern symmetry, regularity, organization, object number, connectivity, size, and shape remain largely unexplored due to the challenges of obtaining such data (*Chan, Stevens & Todd, 2019*; *Glimm et al., 2021*; *Lee, Cavener & Bond, 2018*).

Traditional approaches to obtaining and quantifying pattern variation often rely on human perception of color and patterns (*e.g.*, *Allen et al., 2020*). However, although extremely valuable, a limitation of this method lies in its reliance on typically user-defined categories such as spots, stripes, bands, or other broad classifications (*e.g.*, *Allen et al., 2020*; *Brown & Clegg, 1984*; *Medina, Losos & Mahler, 2016*; *Semler, 1971*; *Tan & Li, 1934*; *Van den Berg et al., 2020*), which are difficult to apply to complex patterns. Additionally, methods quantifying characteristics of the color pattern are still lagging behind (but see *Chan, Stevens & Todd, 2019*; *Glimm et al., 2021*), mostly due to the existence of different software

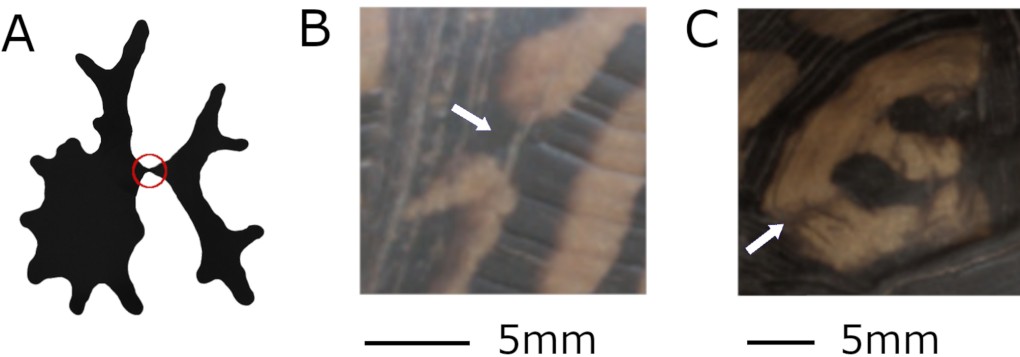

**Figure 2 Defining discrete objects.** When the boundaries between two pattern objects are ambiguous, irregular protrusions may overlap unpredictably when the pattern is extracted, turning two objects into one object or vice versa. The red circle in (A) provides an illustrated example of potential pattern ambiguity, the white arrows in (B) and (C) demonstrate how this occurs in a turtle pattern. (B) The image shows a very thin line that might variably connect two pattern objects (turtle ID mn15, from left view) and (C) shows color variation within a pattern region that might variably separate two pattern objects (turtle ID mn07, from top view).

or packages that allow only certain components of the pattern (*e.g.*, spatial distribution or contrast, general pattern matching, aspect ratio of the objects) to be retrieved or that are optimized for simple, discrete patterns (*e.g.*, *Chan, Stevens & Todd, 2019*; *Hemingson, Cowman & Bellwood, 2024*; *Stoddard, Kilner & Town, 2014*; *Taylor, Gilbert & Reader, 2013*). For simple color patterns, a more detailed analysis can be carried out on the size and shape of the pattern, the orientation and aspect ratio of the objects that compose the pattern, and on the averaged centroid size of the objects (*Chan, Stevens & Todd, 2019*; *Van den Berg et al., 2020*). However, these approaches are less effective when pattern objects are lightly connected, overlapping, or otherwise complex (Figs. 1 and 2).

Together with the challenges of analyzing complex animal color patterns, various factors, including lighting conditions, image capture methods (*e.g.*, camera angle), and the animal's shape or movement (*e.g.*, rounded *vs.* flat bodies) can influence variation in the pattern measurements. Data collection techniques significantly affect measurements of coloration and contrast (*e.g.*, *Akkaynak et al., 2013*; *Johnsen, 2016*; *Schirmer et al., 2023*). Non-uniform lighting in field settings can cause discrepancies between the study subject and the color standard used for calibration, complicating data accuracy (this work; *Lorioux-Chevalier et al., 2023*). Although color standards help standardize images taken under different lighting (*Troscianko & Stevens, 2015*; *Van Belleghem et al., 2018*), variations in lighting across the organism in the same picture can result in overly dark or light areas, reducing pattern accuracy (this work; *Akkaynak et al., 2013*). Furthermore, the impact of image capture methods and organism shape on pattern measurements remains largely unexplored. Identifying which measurements are robust to these variables is crucial for ensuring the reliability of color pattern analyses.

In this work, we use a multi-color threshold approach (segmenting the pixels based on red, green and blue component (RGB) values (*Glimm et al., 2021*; *Van Belleghem et al., 2018*)) to identify and quantify overall color pattern variation in the Eastern box turtle

(*Terrapene carolina*) and assess the influence of several factors related to data capture and analysis on variation in color pattern measurements. Specifically, we investigate how the angle and light at which pictures are taken (angle at which the camera is positioned, controlled *versus* natural light, and where the color standard is placed in respect to the studied organisms/area of interest), how using slightly different threshold values (+/−5%, +/−10%) during the color data extraction step, and the influence of a curved shell resulted in variation at each of the 19 pattern measurements analyzed in this study. We selected the 19 measurements to capture as many components of the color pattern as possible, including developing a new approach to infer how symmetric the color pattern is. The symmetry of the color pattern is of particular interest in ecology and evolution because it is often highly impacted by both camouflage and sexual selection and relevant to understand developmental processes (*Cuthill, Hiby & Lloyd, 2006*; *Enquist & Arak, 1994*). However, symmetry has mostly been studied from a theoretical point of view (*Cuthill, Hiby & Lloyd, 2006*; *Endler & Mappes, 2017*; *Enquist & Arak, 1994*; *Savriama & Klingenberg, 2011*), instead of measuring and quantifying the amount of symmetry in the color pattern as done in this work (see also *Otaki, 2021*).

For the purpose of this study, we captured images from different views of the carapace and obtained distinct measurements of color pattern in the Eastern box turtle (Fig. 1). We selected this species as individuals could easily be encountered in the field and are available in museums, they have a rigid shell—which removes the issue of working with deformable bodies, as this would add another level of complexity in obtaining the data—and show variation in a complex color pattern (Fig. 3). The curvature of the shell is especially challenging as the angle and distance from the animal at which the pictures are taken may affect some aspects of the pattern; for example, the pattern may be rendered more or less elongated depending on the angle at which the image is taken.

Ultimately, this study relies on a simple image capture method that can be applied to other organisms sampled in the field and provides a clear pipeline and MATLAB codes on how to extract color pattern measurements—including some completely new ones, such as color pattern symmetry. We provide guidelines on how to discern which measurements best capture the complexity and variation of the pattern while also being most biologically informative and less sensitive to noise due to sampling variation. Our method for quantifying complex color patterns in box turtles provides tools to further study how selective pressures and different functions of coloration and color patterns like camouflage, thermoregulation, and mate recognition may shape morphological traits, while also offering clues about shell development and adaptive responses to environmental change. Beyond its relevance for understanding turtle biology, this approach can inform broader studies of animal coloration, conservation, biomimetics, and public engagement with biodiversity and evolution.

## MATERIALS AND METHODS

All capture, handling, and experimental protocols were approved by George Mason University IACUC committee (Permit number 1908275). Experiments were carried out to

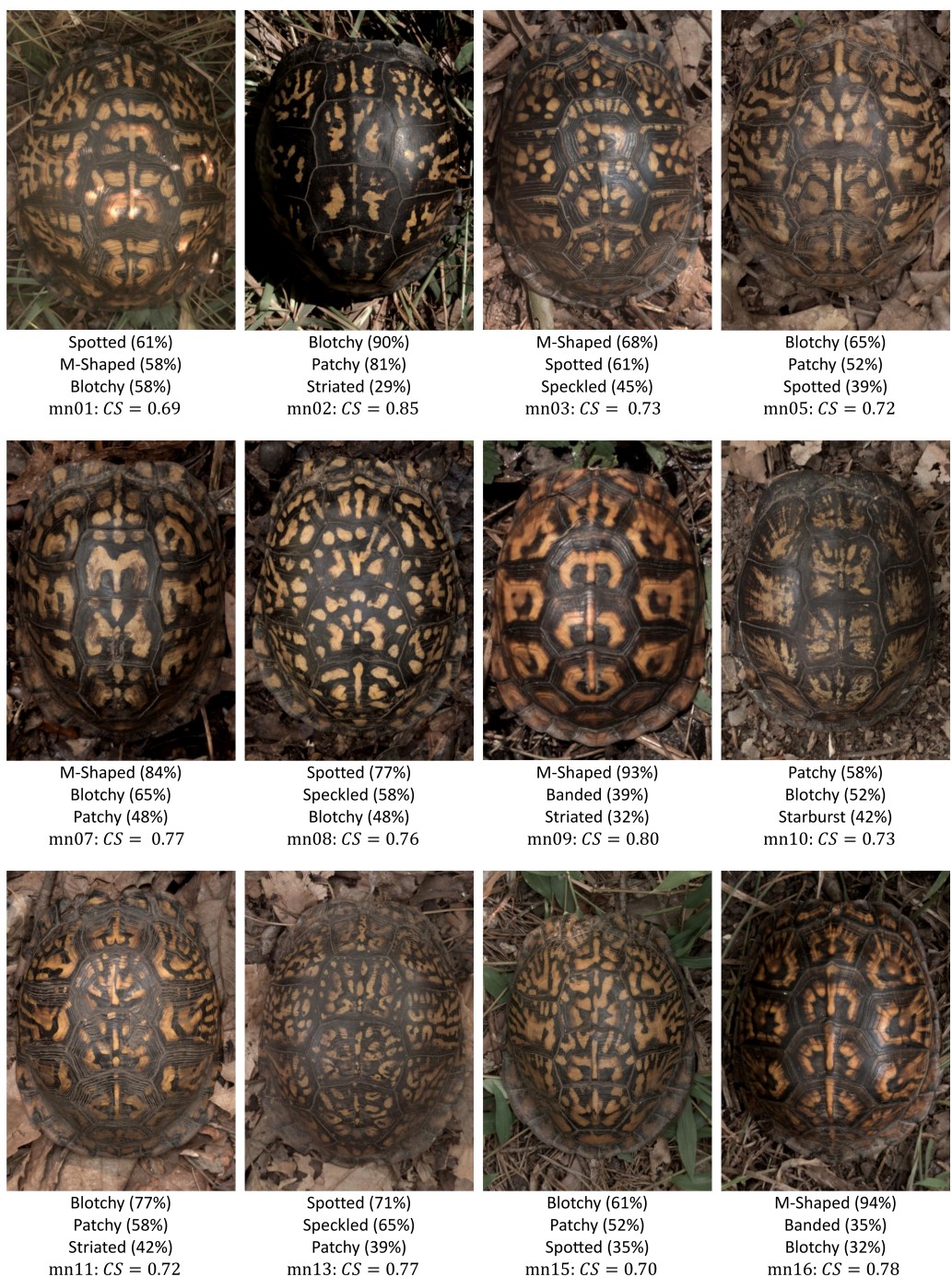

Spotted (61%)
M-Shaped (58%)
Blotchy (58%)
mn01: $CS = 0.69$

Blotchy (90%)
Patchy (81%)
Striated (29%)
mn02: $CS = 0.85$

M-Shaped (68%)
Spotted (61%)
Speckled (45%)
mn03: $CS = 0.73$

Blotchy (65%)
Patchy (52%)
Spotted (39%)
mn05: $CS = 0.72$

M-Shaped (84%)
Blotchy (65%)
Patchy (48%)
mn07: $CS = 0.77$

Spotted (77%)
Speckled (58%)
Blotchy (48%)
mn08: $CS = 0.76$

M-Shaped (93%)
Banded (39%)
Striated (32%)
mn09: $CS = 0.80$

Patchy (58%)
Blotchy (52%)
Starburst (42%)
mn10: $CS = 0.73$

Blotchy (77%)
Patchy (58%)
Striated (42%)
mn11: $CS = 0.72$

Spotted (71%)
Speckled (65%)
Patchy (39%)
mn13: $CS = 0.77$

Blotchy (61%)
Patchy (52%)
Spotted (35%)
mn15: $CS = 0.70$

M-Shaped (94%)
Banded (35%)
Blotchy (32%)
mn16: $CS = 0.78$

**Figure 3** **Variety of turtle patterns found at one of the field sampling localities used in this study (Mason Neck Wildlife Refuge, Virginia, USA).** Included with each turtle pattern are the three most common labels assigned by volunteers, the percentage of volunteers that assigned that label, and the consensus score based on the volunteer responses for all nine label assignments described in 'Categorization and Complexity of Pattern'.

minimize stress and disturbance to the animals and in accordance with relevant guidelines and regulations. Permission was granted to work at the Clifton Institute without the requirement of a permit. United States Fish and Wildlife provided a permit to work at Mason Neck Wildlife Refuge (permit number 51600-22RES05).

## Sample collection

Turtle samples of 55 turtles of the species *Terrapene carolina* were obtained from the Smithsonian Natural History Museum ($n = 43$) and from two nearby field sites known to have populations, Mason Neck Wildlife Refuge and The Clifton Institute (Virginia, USA; $n = 12$ in total). The samples selected from the Smithsonian were from locations in Virginia, Maryland and Washington D.C. All the individuals sampled at the Smithsonian Natural History Museum were preserved in ethanol. Field sampling was carried out to test for the influence of varying light conditions on the effectiveness of the color pattern capture method used in this work for animals sampled in the wild. In the field, searches were conducted early in the morning or later in the afternoon when temperatures were around 26° Celsius and 66% humidity (optimal average temperature and humidity to encounter box turtles in this area; E Maki, pers. obs., 2023). For this study we selected only individuals with a relative length >80 mm and a carapace width/length ratio <90%, since smaller individuals do not show the complexity of the pattern (E Maki, pers. obs., 2023). Younger turtles tend to be rounder in shape with a width measurement very close to the length measurement, adult length increases significantly more than their width as they grow (*Adamovicz et al., 2018*; *Langtimm, Dodd & Franz, 1996*; *Way Rose & Allender, 2011*). Males and females were identified based on the sexual dimorphism of the plastron, in which males have a slightly more concave plastron than females (*Elghammer et al., 1979*; *Yahner, 1974*; *Biewer et al., 2024*). Measurements were all taken in the field with a digital caliper.

## Data collection
### Turtle photographs

Photography took place in both controlled (museum) and field environments, but the methods used for taking the images were the same in order to compare the results obtained for the two different sampling conditions. All Smithsonian specimens were removed from the ethanol, dried slightly, and placed in a white bin next to an 18% gray color standard calibration card (brand: Digital Grey Kard) cut to 50 mm X 30 mm dimensions (Fig. 4). The gray color standard card was also placed next to the turtles sampled in the field (Fig. 4). As box turtles retract their head and stay still during encounters with humans, animals sampled in the field were always still. For each sampled animal (in the field or in the museum), photos were taken from five different viewpoints in order to capture variation in pattern across the entire carapace: top, front, back, left and right views (Fig. 4). Our approach is based on the idea that different body regions may experience distinct selective pressures (*e.g.*, *Allen et al., 2020*; *Glimm et al., 2021*). Based on our previous work (*Glimm et al., 2021*), we hypothesize that different views of the shell of box turtles may show varying levels of morphological variation. Images of each turtle encountered in the field were taken at the time of encounter and in the location where the animal was found. The turtle shells

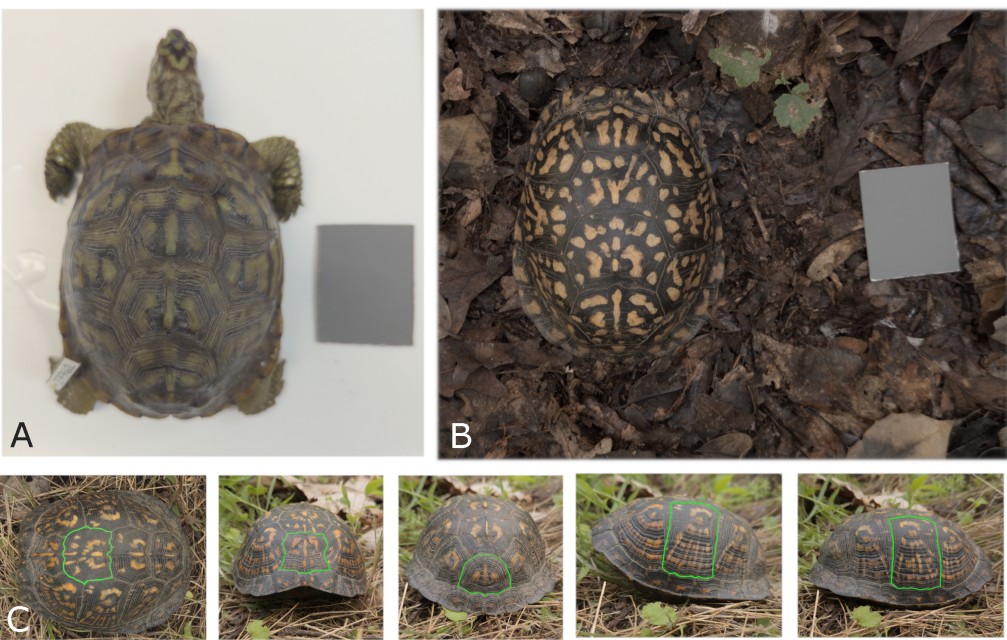

**Figure 4** **Photographing conditions and viewpoints used for turtle images.** (A) A museum specimen from the Smithsonian collection (ID 27761) photographed in a white bin with an 18% gray standard for calibration. (B) A Mason Neck field turtle (ID mn08) photographed at the site of capture, with the 18% gray standard placed next to the animal. Some distortion of the pattern due to shell curvature is visible in this top view, particularly for the side scutes. The effect of distortion caused by the viewing angle is evaluated in 'Influence of the angle at which pictures are taken'. (C) Diagram of the five standardized viewpoints used for all turtles: top, front, back, left, and right. The turtle shown here is a Mason Neck turtle (ID mn20). With digital segmentation, a scute (scale of the shell) is outlined in a unique color and isolated from the image for analysis. Each view has one scute digitally segmented.

were not cleaned of dirt or dust, though they did not have so much dirt as to occlude the pattern. All photos were taken in RAW format (.dng for the camera used) using a Google pixel 6 cellphone camera set to the default settings as follows: 1.2 µm pixel width, f/1.85 aperture, 82-degree field of view and 1/1.31 image sensor size. Each photo was taken from a distance of about 30 cm from the turtle using the natural light in the field or the fluorescent lighting at the Smithsonian for each specimen. The cellphone was held by hand without any tripod or holding device, except for images used for studying the influence of the angle at which images are taken (see below). One single person obtained the photos for all the individuals included in this study. For each view, two photos (for a total of 10 photos per turtle) were taken placing the camera as parallel to the view as possible. Since photos were obtained without holding devices and the distance from the animal was therefore a rough estimate, photos of each view were taken in duplicates to estimate the influence of the photo capture on variation in the studied measurements. Each duplicate photo was taken immediately after the initial photo without moving or adjusting the turtles. We tried to also maintain the camera position between the two pictures invariant.

For field-sampled animals, images often need to be taken without tripods or holding devices, and animal movement during photography can cause variations in the camera's

distance and angle relative to the subject. To estimate the influence of non-standardized image angle between the turtle and the camera on variation in the pattern measurements, the first three turtles encountered in the field on one random sampling day had images taken at different angles with three images for each view of each individual (instead of two as described above). To ensure that the different angles at which the images were taken were exactly the same across the tested individuals, the cellphone used to take the pictures was placed on a tripod (Amazon Basics) and a phone holder (SharingMoment) at 60 cm straight above each view of the turtle. To obtain different angles at which the photos were taken, each turtle was tilted using a protractor (Westcott). Photos were taken at 0-, 5-, and 10-degrees angles to simulate realistic variation that could happen when taking images of an animal without standardized camera placement. Angles were measured on the turtle by lining up the lowest edge of the carapace (around the middle point of the turtle) with the desired angle of the protractor. Turtles were then tilted to the left (-5, -10), right (+5, +10) and straight overhead (0) for the top view (five images in total for the top view, in triplicate). Turtles were tilted towards the camera (+5, +10) and (0) (camera on the ground parallel to the viewpoint) for the front, back, left, and right views (three images in total for the front, back, left and right views, in triplicate) (Fig. 5). When the camera is positioned as close to the ground as possible for the front, back, left, and right views, it limits the tilt to one direction. However, due to the nature of shooting from above, the camera is more likely to tilt to the left or right in the top view. Zero degrees were obtained when the photo was taken with the camera parallel to the top view of the turtle.

Finally, as organisms sampled in the field may have parts exposed to different light conditions, the gray standard card used for color calibration may not accurately reflect this variation. As such, we wanted to infer the influence of different lighting conditions between the gray standard calibration card and the study object. To do this a basketball hat was placed under three different lighting conditions (full shade, partial shade and full sun) and three photos each from the top and from the front were taken with an 18% color standard card being placed at different distances and lighting conditions from the hat (Fig. S1).

### Image processing

Photos were calibrated using the 18% gray color standard card included in each picture (Figs. 4A and 4B). Turtles were not segmented from the background and the entire picture was color calibrated. Color calibration was done in order to correct color differences among images due to different lighting conditions. The Multispectral Image Calibration and Analysis (MICA) toolbox through ImageJ (*Troscianko & Stevens, 2015*) was used to color calibrate each photo following the program guidelines. The gray color standard was identified by dragging a box around the visible calibration card in the image. Field images of the turtle and the color standard card often had variable lighting conditions. Ideally, lighting conditions are uniform across the image and the card, and any sub-section of the card can be used for calibration (it is not necessary to use the entire card since it is of uniform value). In the case where lighting conditions were not uniform across the card due to shadows, a sub-section of the card illuminated similarly to that of the turtle was selected. After calibration, photos were then converted lossless from .dng to .png for use

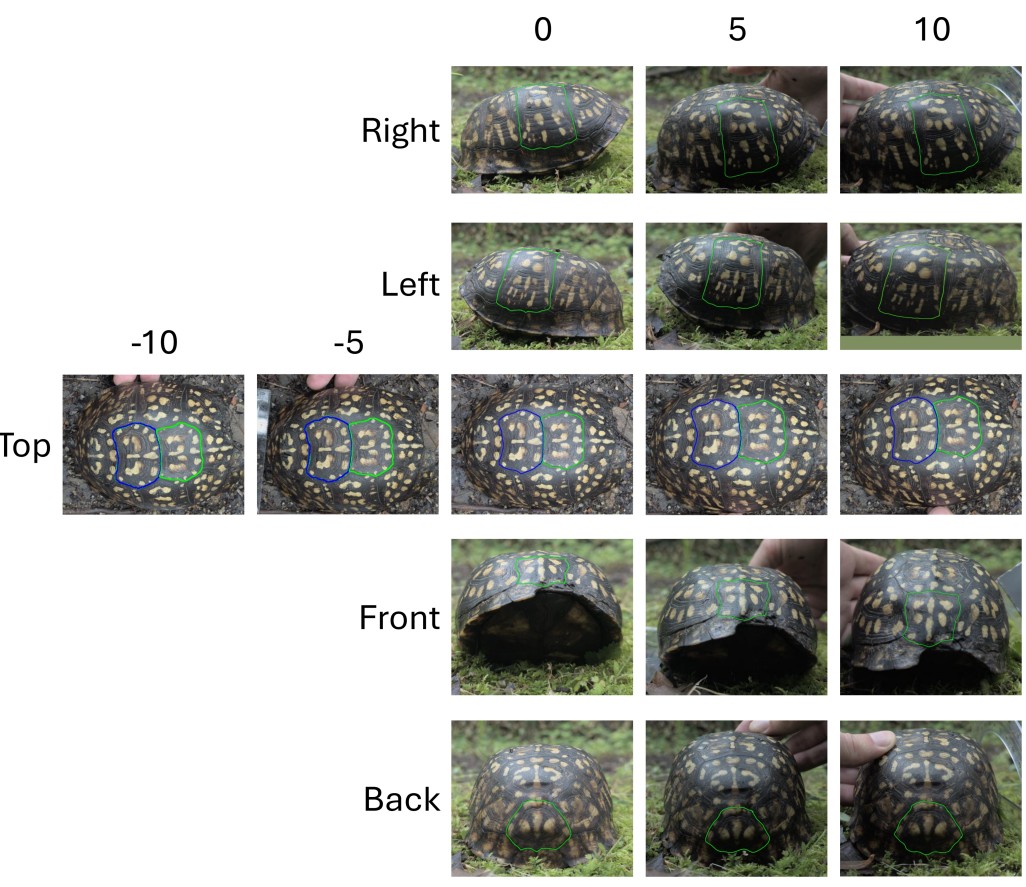

**Figure 5   Viewpoints and camera angle variations used to test measurement noise.** All viewpoints (right, left, top, front and back) were used to assess the measurement noise introduced by the imaging process. For the right, left, front and back, three angles were tested (0°, 5° and 10°). For the top viewpoint, five angles were tested (−10°, −5°, 0°, 5°, 10°).

in the GNU Image Manipulation Program (GIMP) (*The GIMP Development Team, 2019*) for the next steps. The images of the red bill of the cap were processed using the same method and algorithm as the turtle images, applying hue, saturation and brightness (HSB) measures to estimate the associated measurement noise.

Since there was not a clear standardized distance used for taking the photographs, each image was on a different scale. The image length scale was then based on the number of pixels per millimeter, by measuring the number of pixels along the 50 mm edge of the gray standard card in GIMP.

The turtle color pattern in this study was defined as any coloration that was more yellow than average in each scute. The criterion for selecting coloration more yellow than average was based on an initial visual inspection of the species' color pattern (typically shades of yellow or orange) conducted by the authors. Color pattern information was therefore extracted on a scute-by-scute basis. For this study, for each individual, we selected only the five scutes that were least affected by the curvature of the carapace for each view in order to avoid pattern distortion effects: top middle, left middle, right middle, front bottom
middle and back bottom middle (Fig. 4C). Although the 3D shape of the shell is of primary importance when studying how predators or conspecific would see the color pattern, accurately reconstructing the pattern across the 3D surface would require reconstructing the shell in 3D (see for example *Chiari et al., 2008*; *Chiari & Claude, 2011*) which may be the goal of future work.

Each of the selected five scutes of each individual was manually outlined following the scute boundaries (Fig. 4C) using GIMP. For outlining, the color RGB (0,255,0), a very bright green, was used. The outline color matters in that it is uncommon to be found naturally in the image. Any colors can be chosen as long as they are not present within the rest of the image. Outlines were as precise as possible along the borders of the scutes, as imprecise outlines may cause variation (*e.g.*, by including parts of the objects of the pattern of another scute for example). Scute boundaries can be identified by the well-defined "lines" that separate neighbor scutes (Figs. 4 and 5).

### Color pattern identification algorithm

A pattern recognition algorithm was developed in MATLAB (*The MathWorks Inc, 2020*) to extract pattern objects that were "more yellow" than average within each scute, defined by a threshold red-minus-blue difference. This criterion was chosen as yellow primarily consists of red and green light, excluding blue.

The pattern recognition algorithm was designed to identify the color pattern of each image in a fully automated way. It begins with a set of color- and light-calibrated images, each containing a digitally segmented scute. The algorithm processes each image independently and automatically, producing a corresponding set of binary images in which each pixel within the scute is classified as either pattern or background. The steps for the pattern extraction are described below (steps 1–4) and illustrated for two scutes in Fig. S2.

### Step 1: Dynamic threshold for yellow detection

In the first pattern extraction step, we identified a "pre-pattern" as the set of pixels in each scute that were a threshold level more yellow than average for that scute. The threshold level is dynamic and adjusts to the overall yellow of each image because the average amount of yellow in an image varied due to lighting conditions, and also the yellow contrast between the pattern background and within the pattern foreground varied substantially from turtle to turtle.

The threshold yellow level was estimated using the red-minus-blue channel difference:

- the red-minus-blue value was calculated at each pixel as the difference of the red and green RGB channel values at that pixel: $R - B$
- The average red-minus-blue value was calculated for all pixels within the scute: $mean(R - B)$
- the threshold yellow level was 110% of the average red-minus-blue value (a scute pixel is marked as a pattern pixel if $R - B > (1.1 \times mean(R - B))$ at that pixel)

The average red-minus-blue value is unique to each image scute. By trial and error, the threshold parameter of 110% of each scute's average red-minus-blue difference was determined to most closely capture the yellow pattern identified by eye. We test the effect

of varying this threshold parameter on the captured pattern by measuring the sensitivity of pattern measures to the threshold in 'Influence of the choice of threshold value'.

### Step 2: Removal of spurious small objects

Once the set of *pixels* that were more yellow was identified by the formula described above, isolated pixels and groupings of pixels less than the threshold size of one mm$^2$ were removed. Depending on the spatial resolution of the image (*i.e.,* pixels per mm), this threshold varied from 25 to 400 pixels. These pixels were identified as more yellow than average by the previous step, but were likely to be noise (or spurious non-pattern objects) since they were small and disconnected from larger yellow regions.

### Step 3: Refinement of detected patterns

The third step of the pattern cleaning process was to fill small "holes" found within the identified yellow regions. Within connected yellow regions, isolated pixels often failed to meet the yellow threshold, possibly due to noise such as light scattering or glare. Gaps in the pattern objects were filled if they were smaller than one mm$^2$.

### Step 4: Smoothing pattern edges

In a final fourth step, the edges of the yellow pattern were then smoothed, again with respect to the length scale of the image so that thin protrusions were removed, and contour fluctuations were smoothed and averaged at a fixed spatial scale for all scutes. This was achieved by having the boundary of the pattern eroded and then dilated with a 0.5 mm x 0.5 mm square structuring object. Erosion removes small amounts of the edge pixels of the pattern then dilation refills the pixels which results in a net zero effect except to remove thin and small size noise of the pattern along the edges. As an example, there may be a one-pixel wide (or otherwise thin) line that connects two pattern objects making it one object instead of two (Fig. 2B). The erosion and dilation steps will remove a very thin line connecting objects without changing the size of the objects.

The full pipeline for the analyzing digital images included MICA and GIMP to initially color calibrate the photos (see 'Image processing'), the pattern was extracted in MATLAB as described here, the measures were computed in MATLAB (see 'Pattern measurements'), and the analysis of the measurement data (*e.g.*, CV calculations) was completed in Excel and R.

### Pattern measurements

After the color pattern extraction steps, 19 pattern measurements were obtained (Table 1). These measurements were selected to quantify as many aspects of the color pattern as possible, including measures quantifying the general size, shape and number of objects in the pattern (E, PA, Ob, OA), color (H, S, B), contrast between the pattern and its background (ED, IC, RC, GC, BC, YC), and overall pattern distribution and organization (FA, PL, Sy, CR, OF, NO). Table 1 includes the descriptions of each measurement with their abbreviations and how they are calculated.

**Table 1  Description of the 19 pattern measurements used in this work.** For each measure, two examples are shown of relatively low and high measure values with the corresponding turtle pattern. *The convex hull is the smallest convex set that encloses all the pixels of the pattern, forming a convex polygon. In MATLAB, this is computed using bwconvhull.

| Name | Definition and Formula | Purpose | Examples with low and high values among study turtles |
|---|---|---|---|
| **Fractional area (FA)** | The fractional area is calculated as the total number of yellow image pixels divided by the total number of pixels in the scute region: $FA = \frac{\Sigma(pattern\ pixels)}{\Sigma(scute\ pixels)}$ | Provides a measure of the fraction of space the pattern occupies in the scute. Based on the binary pattern image. |  Turtle 497203 Avg FA = 0.2   Turtle 22468 Avg FA = 0.38 |
| **Mean eccentricity (E)** | The eccentricity of the pattern objects is a value between 0 (a perfect circle) and 1 (a perfect line). It is calculated using stats.Eccentricity of the *regionprops* MATLAB subroutine as the ellipticity of the ellipse with the same second moments as the object. For an ellipse with major axis $a$ and minor axis $b$, the ellipticity is $\sqrt{1 - b^2/a^2}$. | Quantifies the mean shape of the region enclosing each of the pattern objects in a scute. Based on the binary pattern image. |  Turtle mn8 Avg E = 0.76   Turtle 497276 Avg E = 0.88 |
| **Peak length (PL)** | The peak length is the average distance between pattern objects (*Miura, Komori & Shiota, 2000*) computed by finding the skeletonization of the positive and negative of each image (valleys and peaks, respectively): $PL = \frac{2 \cdot \Sigma(scute\ pixels)}{(\Sigma valley\ pixels) + (\Sigma peak\ pixels)} \cdot mm/pixel$ | Provides a measure of the characteristic length scale of the image, roughly corresponding to the spacing of typical pattern objects. Based on the binary pattern image. |  Turtle mn16 Avg PL = 4.27   Turtle 203309 Avg PL = 7.98 |
| **Perimeter/Area (PA)** | The ratio of the perimeter and the area is the mean perimeter divided by the mean area of the pattern objects where the perimeter and area of objects is calculated using stats.Perimeter and stats.area of the *regionprops* MATLAB subroutine. $PA = \frac{mean\ object\ perimeter}{mean\ object\ area} \cdot mm/pixel$ | Quantifies aspects of the shape and area of the pattern objects. Larger wider patterns should have a lower value and longer and skinnier patterns have higher values. Based on the binary pattern image. |  Turtle 22468 Avg PA = 0.6   Turtle 519611 Avg PA = 1.4 |
| **Hue (H)** | The hue measures the color component of the pattern and has values that range from 0 to 1. For example, 0.00 = red, 0.33 = green, and 0.66 = blue. The image is converted from RGB to HSB in MATLAB and the hue is calculated as the mean of the hue values of all the pixels identified as pattern pixels. | Provides a single value that can easily quantify the color of the pattern. Based on the HSB of the image. Based on the HSB pixel color channels. |  Turtle 288185 Avg H = 0.06   Turtle mn8 Avg H = 0.17 |

**Table 1** (*continued*)

| Name | Definition and Formula | Purpose | Examples with low and high values among study turtles |
|---|---|---|---|
| **Saturation (S)** | The saturation measures the color intensity of the pattern with values ranging from 0 (indicates no saturation or grayscale ) and 1 (indicates full saturation). The image is converted from RGB to HSB in MATLAB and then saturation is calculated as the mean of the saturation values of all the pixels identified as pattern pixels. | Provides a measure for which the overall saturation or intensity of the pattern color can be quantified in a single measurement. Based on the HSB pixel color channels. | <br>Turtle mn10 Avg S = 0.17  Turtle 49607 Avg S = 0.64 |
| **Brightness (B)** | The brightness of the pattern ranges from 0.0 (indicates very dark or black) to 1. 0 (indicates very bright or white color). The image is converted from RGB to HSB in MATLAB and then brightness is calculated as the mean of the brightness values of all the pixels identified as pattern pixels. | Provides a measure of the brightness of the color pattern. This is highly influenced by the amount of environmental light (Troscianko & Stevens, 2015). Based on the HSB pixel color channels. | <br>Turtle mn15 Avg B = 0.17  Turtle 22612 Avg B = 0.49 |
| **Symmetry (Sy)** | Symmetry is measured as the ratio of pixel overlap when a mirror-image copy of the pattern is overlaid onto the original image and rotated if needed. The symmetry index Sy is computed as the maximum pixel overlap over all rigid transformations (translations + rotations) of the mirror image copy. This algorithm was implemented in MATLAB via a brute-force method that searches over a discretization of all rigid transformations. | Provides a value between 0 and 1 that quantifies the axial symmetry of the pattern with respect to the center axis of the scute. Provides information about the mirror symmetry of the left and right half of the pattern. Based on the binary pattern image. | <br>Turtle 22609 Avg S = 0.53  Turtle 497302 Avg S = 0.81 |
| **Euclidean distance of pixel color (ED)** | The Euclidean distance measures the average distance between RGB pixel values from the identified pattern (pixels more yellow than average) and non-pattern (dark background of the scute). $ED = \sqrt{(RC)^2 + (GC)^2 + (BC)^2}$ | Provides a single value that can quantify the contrast between the average values of the RGB between the pattern and non-pattern. Based on the RGB pixel color channels. | <br>Turtle 288186 Avg ED = 27.84  Turtle 203066 Avg ED = 94.29 |
| **Intensity contrast (IC)** | The intensity contrast was measured for the RGB image converted to gray scale as the difference of the mean intensity of the pattern (pixels more yellow than average) and non-pattern (dark background of the scute) pixels. IC = Mean Pattern Intensity − Mean Non-Pattern Intensity | Provides a measure that quantifies the difference in intensity between the pattern and non-pattern. Based on the grayscale image. | <br>Turtle 288186 Avg IC = 16.15  Turtle 42825 Avg IC = 49.17 |

**Table 1** (*continued*)

| Name | Definition and Formula | Purpose | Examples with low and high values among study turtles |
|---|---|---|---|
| **Number of Objects (Ob)** | The number of objects is the total number of objects composing the pattern identified by the pattern recognition algorithm. In the MATLAB subroutine, *bwlabel* was used to number the objects. $Ob = number\ of\ objects$ | Quantifies the number of objects in each pattern. Provides a number that can determine how interconnected the patterns are. It can also be used as a building block for other measurements. Based on the binary pattern image. |  Turtle 497302 Avg Ob = 2.6   Turtle 22609 Avg Ob = 19.39 |
| **Average object area (OA)** | The average object area was calculated as the mean of the area for each of the identified pattern objects. The area of each object was calculated using stats.Area of the *regionprops* MATLAB subroutine. $OA = mean(object\ area) \cdot (mm/pixel)^2$ | Provides a mean measure of the size of the objects composing the color pattern. Quantifies the average overall size of the pattern objects. Based on the binary pattern image. |  Turtle 519611 Avg OA = 16.66   Turtle 22468 Avg OA = 131.37 |
| **Red contrast (RC)** | Difference of the average red pixels of the identified pattern (pixels more yellow than average) versus the non-pattern (dark background) pixels, where the red value of a pixel was the value of the red RGB channel. $$RC = mean\begin{pmatrix} red\ of \\ pattern \\ pixels \end{pmatrix} - mean\begin{pmatrix} red\ of \\ non-pattern \\ pixels \end{pmatrix}$$ | Provides a quantification of contrast present in the red pixels between the identified pattern and the non-pattern (dark background of the scute). Based on the RGB pixel color channels. |  Turtle 288186 Avg RC = 22.56   Turtle 42825 Avg RC = 63.86 |
| **Blue contrast (BC)** | Difference of the average blue pixels of the identified pattern (pixels more yellow than average) versus the non-pattern (dark background) pixels, where the blue value of a pixel was the value of the blue RGB channel. $$BC = mean\begin{pmatrix} blue\ of \\ pattern \\ pixels \end{pmatrix} - mean\begin{pmatrix} blue\ of \\ non-pattern \\ pixels \end{pmatrix}$$ | Provides a quantification of contrast present in the blue pixels between the identified pattern and the non-pattern (dark background of the scute). Based on the RGB pixel color channels. |  Turtle 288210 Avg BC = 6.15   Turtle 203066 Avg BC = 32.32 |

**Table 1** (*continued*)

| Name | Definition and Formula | Purpose | Examples with low and high values among study turtles |
|---|---|---|---|
| **Green contrast (GC)** | Difference of the average green pixels of the identified pattern (pixels more yellow than average) versus the non-pattern (dark background) pixels, where the green value of a pixel was the value of the green RGB channel. $$GC = mean\begin{pmatrix} green\ of \\ pattern \\ pixels \end{pmatrix} - mean\begin{pmatrix} green\ of \\ non-pattern \\ pixels \end{pmatrix}$$ | Provides a quantification of contrast present in the green pixels between the identified pattern and the non-pattern (dark background of the scute). Based on the RGB pixel color channels. | Turtle 22613 Avg GC = 13.18 — Turtle mn8 Avg GC = 48.77 |
| **Yellow contrast (YC)** | Difference of the average yellow pixels of the identified pattern (pixels more yellow than average) versus the non-pattern (dark background) pixels, where the yellow value of a pixel was defined as the minimum of the red and green RGB pixel values: $$YC = mean\begin{pmatrix} yellow\ of \\ pattern \\ pixels \end{pmatrix} - mean\begin{pmatrix} yellow\ of \\ non-pattern \\ pixels \end{pmatrix}$$ | Provides a quantification of contrast present in the yellow pixels (combination red and green pixels) between the identified pattern and the non-pattern (dark background of the scute). Based on the RGB pixel color channels. | Turtle 288210 Avg YC = 6.18 — Turtle mn8 Avg YC = 34.08 |
| **Centrality ratio (CR)** | The centrality ratio is the ratio of the average distance of pattern versus non-pattern pixels from the center of the scute, where the center is the centroid $(Cx, Cy)$. $$CR = \frac{mean\begin{pmatrix} pattern\ pixel \\ distances\ to\ centroid \end{pmatrix}}{mean\begin{pmatrix} non-pattern\ pixel \\ distances\ to\ centroid \end{pmatrix}}$$ | Provides a measure to quantify on average how close the pattern pixels are to the center of the scute in comparison to the non-patten pixels. Based on the binary pattern image. | Turtle 497261 Avg CR = 0.63 — Turtle 519611 Avg CR = 1.11 |
| **Occupation factor (OF)** | The number of pixels of the pixelated convex hull* is divided by the total number of scute pixels. $$OF = \frac{(\Sigma(Convex\ Hull))}{(\Sigma(scute))}$$ | Provides another measure to quantify the fraction of the scute area that the pattern covers in comparison to the overall pixels, but specifically focuses on the convex hull of the pattern. Based on the binary pattern image. | Turtle 497261 Avg OF = 0.45 — Turtle 22609 Avg OF = 0.93 |

**Table 1** (*continued*)

| Name | Definition and Formula | Purpose | Examples with low and high values among study turtles |
|---|---|---|---|
| **Normalized offset (NO)** | The normalized offset is measured as the distance of the pattern center from the scute center normalized by the scute radius which is computed as the square root of the scute area. $$NO = \frac{\sqrt{(Px-Cx)^2+(Py-Cy)^2}}{\sqrt{\Sigma(scute\ pixels)}},$$ where $(Px, Py)$ is the centroid of the pattern pixels and $(Cx, Cy)$ is the centroid of the scute pixels. | Provides a measure to quantify how far off center the pattern is in relation to the center of the scute. Based on the binary pattern image. |  Turtle mn15 Avg NO = 0.03    Turtle 497203 Avg NO = 0.19 |

## Measurement noise analysis

In this study, we investigate how various factors, such as the angle at which photos are taken, different images of the same subject, and the placement of the color calibration standard card influence the variation in pattern measurements. Regardless of the factor being tested, the influence of these factors was assessed using the coefficient of variation (CV) of pairwise differences. The CV in this study corresponds to the pairwise difference of the image measurements obtained from the two (or three for the angle testing) pictures of the same view on the same animal. CVs were calculated for each measurement separately. CV calculated for three pictures per view used the standard CV formula $\frac{\sigma}{\mu}$, where $\sigma$ is the standard deviation and $\mu$ is the mean of the measurements. With only two images, however, the standard deviation underestimates variability. The pairwise difference divided by the mean offers a more accurate measure of relative variability between the two values. For consistency, we refer to this relative difference calculation as ''CV'' in two-image comparisons, which was computed using this equation:

$$Pairwise\ Difference = CV = \left| \frac{image\ 1 - image\ 2}{mean\ (image\ 1\ \&\ 2)} \right|.$$

This analysis allows inferring the robustness of each pattern measurement under the different tested sampling conditions. Coefficients of variation above 10% were considered to indicate substantial variation in that measurement.

A $p$-value was computed to test whether pattern measurements differed significantly between images from the field and those from the museum ('Variation in pattern measurements due to sample source'), and again to assess whether the coefficient of variation (CV) of any measurement was significantly correlated with the consensus score ('Categorization and complexity of pattern'). Because $T$-tests, $F$-tests, and CV correlations were conducted independently across 19 measurements, $p$-values were adjusted using the Benjamini–Hochberg procedure to control the false discovery rate.

The absence of standardized lighting for field turtles could greatly affect the hue, saturation and brightness (HSB) (Table 1, Table S1) if the standard is not exposed to the same lighting conditions as the turtle, leading to variations in HSB between photographs (Table S1). To test the influence of the placement of the gray standard in comparison to

the object of interest, we used images taken of a hat with the gray color standard placed at different distances from the hat. Images (two viewpoints and three photos taken per viewpoint) were then color calibrated following the same methods described above and values of HSB were measured on the images to infer the influence of different lighting between color standard and object of interest on HSB using CVs as described above.

To test the effect of noise in the angle of photography, we tested angles of −5, −10, 0, 5 and 10 degrees from baseline to measure the coefficient of variation as the angle is varied (Fig. 5).

### Sensitivity analysis and threshold selection

The pattern identification algorithm uses a non-uniform threshold where values vary from image to image. We found that a (110%) threshold on the minimum red and blue difference (see 'Color pattern identification algorithm') created the most accurate pattern recognition results based on human visual comparison (Fig. 6). This threshold can be adjusted to add more or less total area ("thickness") to the pattern as it increases or decreases the number of pixels identified as "more yellow" than average (Fig. 6). To test the influence of the threshold on the pattern identified by the algorithm, we changed the threshold from 110% by +/−5% (115% and 105%) and +/−20% (130% and 90%) (Fig. 6). These specific percent increases and decreases were chosen to demonstrate how much the fractional area changes with small and large changes to the threshold. We chose to focus on fractional areas since we found it to be the least sensitive measurement to variation (see 'Results'). As such, although this is a conservative measure, if a threshold was found to affect this measurement, it could be assumed that it would affect all the other measurements even more. Each threshold was then run across all the images in the study. Fractional area measures were then analyzed using CV to determine the variation between the original threshold of 110% and each of the four test thresholds (90%, 105%, 115%, 130%) and the percent change between the test thresholds and the optimal thresholds, generating a framework of how sensitive to changes our algorithm is.

## Citizen science: color pattern categorization to describe complexity

The color calibrated top view of 98 individuals was presented to 31 volunteers of different genders and similar age to obtain an estimate of the color pattern complexity as seen by the human eye. Color pattern complexity was inferred based on the consensus value obtained across the volunteers on categorization of the observed patterns. The volunteers were all undergraduate students at George Mason University who responded to an advertisement asking for participants for a project to categorize the color pattern of different individuals of box turtles. No remuneration or any other benefits was offered to participants and no information on the participants was collected. Images (one top view image per individual) were uploaded to a google drive folder where participants could see but not edit the images. An Excel file with the individual/picture ID listed was also available on the Google drive. Volunteers were instructed to work alone (no volunteer knew the identity of others), to download the Excel file on their computer, and choose up to four categories per individual of the nine categories presented (Table 2). In a pilot study, initial categories were suggested

|  | Top | Left | Right | Front | Back |
|---|---|---|---|---|---|

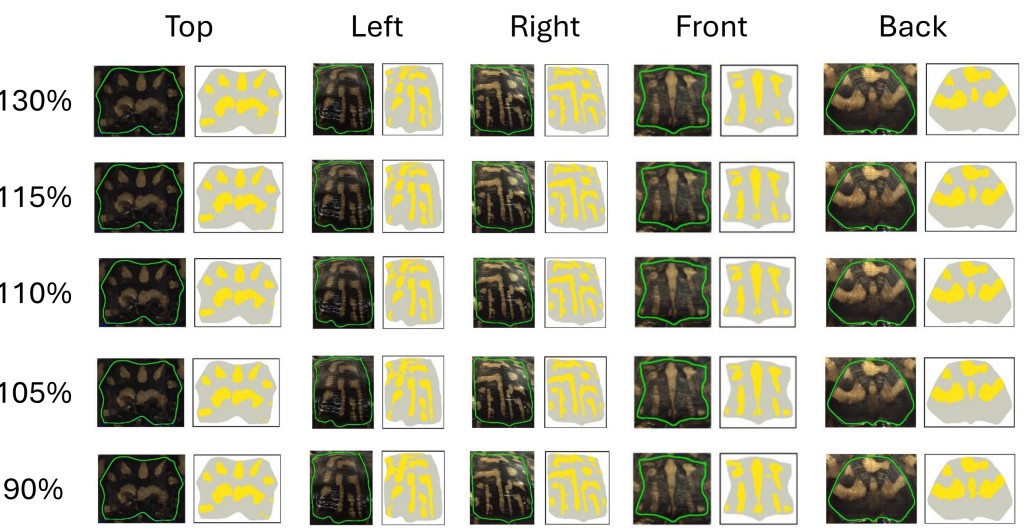

**Figure 6** **Effect of variation of the threshold on the extracted pattern.** Visual comparison between the pattern extracted by the algorithm and as seen by the human eye for each view for the optimal threshold (110%), and the +/-5% (105% and 115%) and +/-20% (90% and 130%) thresholds.

**Table 2** **Categories and descriptions of categories presented to students for the pattern consensus score.**

| Categories | Descriptions |
|---|---|
| **M-shaped** | Defined M shape present in pattern |
| **Single spotted** | Single spot in the middle of the scute |
| **Striated** | Irregular shaped stripes |
| **Blotchy** | Irregular shaped spots |
| **Spotted** | Distinct spots throughout the pattern |
| **Speckled** | Many small spots throughout the pattern |
| **Banded** | Clear uniform striped pattern |
| **Patchy** | Random irregular shaped pattern objects |
| **Starburst** | Pattern radiates out from a clear center point |

to 10 volunteers, and respondents could also add their own pattern categories if they felt that a better classification could be found for a certain pattern. For the follow-up analysis, the most frequent categories write-ins were added and unused categories were subtracted resulting in the nine categories listed in Table 2. Volunteers looked at the images on their own devices (cell phones or computers) and were instructed to carry out the categorization only on the basis of the pattern. Volunteers were provided with the descriptions for each pattern as in Table 2. Although the use of different devices from volunteer to volunteer may affect some aspects of coloration and as such some aspects of the patterns may be more or less visible, we asked volunteers to focus on broad pattern categorization.

Consensus values were then obtained for the pattern of each turtle by calculating a consensus score based on the diversity (low consensus) or uniformity (high consensus) of the categories selected by volunteers. We first quantified the consensus for each category

for each turtle using a consensus proportion method (also known as a majority voting method) by determining whether students more frequently assigned a 0 (category does not apply) or a 1 (category does apply) for that category, with the proportion reflecting the level of agreement among students for that category and that turtle. For example, if six students assigned a '0' and four students assigned a '1' for a category for a single turtle, the consensus for that category was 60%. For a binary assignment as we used, the proportion of the most frequent assignment (the proportion of '0's or '1's, whichever is higher) is a value from 0.5 to 1.0 (low to high category consensus). Since volunteers could choose up to four categories, and the categories were not mutually exclusive, the consensus score was computed independently for each category in this way. Then, the overall consensus score across all categories, for each turtle, was computed as the arithmetic mean of the nine individual consensus proportions for each category. A complete lack of consensus (for example, a random volunteer assignment of '0's and '1's) would predict an overall consensus close to 0.5 and complete consensus (every volunteer agrees on whether a category applies for every category) would have an overall consensus score of 1.0.

The consensus values were used as an indication of complexity of the pattern. The stronger the consensus, the less complex the color pattern was considered. To infer which pattern measurements may best reflect the complexity of the pattern, we ran a correlation analysis using the corresponding value of each pattern measurement (as the average over the two ($n = 2$) top view images) for each turtle *versus* the consensus score. The correlation test was run for the turtle images for which we had both consensus scores and statistical measures, a total of 53 turtle images ($n = 43$ from the Smithsonian Natural History Museum and $n = 10$ from field sites). Although we had collected statistical measures for 55 turtles (see 'Sample collection'), the 98 top views presented to the volunteers inadvertently excluded two of these individuals so that there were only 53 turtles for which we had both consensus scores and statistical measures. The correlation test was run for a total of 19 statistical measures in R (*R Core Team, 2024*) using the corr.test function (Pearson's correlation).

# RESULTS

Our final dataset for the analysis of variation in pattern measurements includes 55 animals each with photos from five views and two-three pictures taken per view (three pictures per view were taken only for the angle analysis) for a total of 610 pictures analyzed (610 = 43*5*2+12*5*3).

## Variation in pattern measurements

We developed algorithms to identify the color pattern and compute the pattern measurements describing the different aspects of the pattern. We tested which of the pattern measurements are more or less variable within and across individuals (3.1) and more or less strongly influenced by variation in sampling conditions (3.2, 3.4, 3.5) or algorithm threshold (3.3). We also measure how variation depends on pattern characterization (3.6). As an estimate of the measurement noise, we use the coefficient of variation (CV) calculated on multiple images of the same view for the same individual.

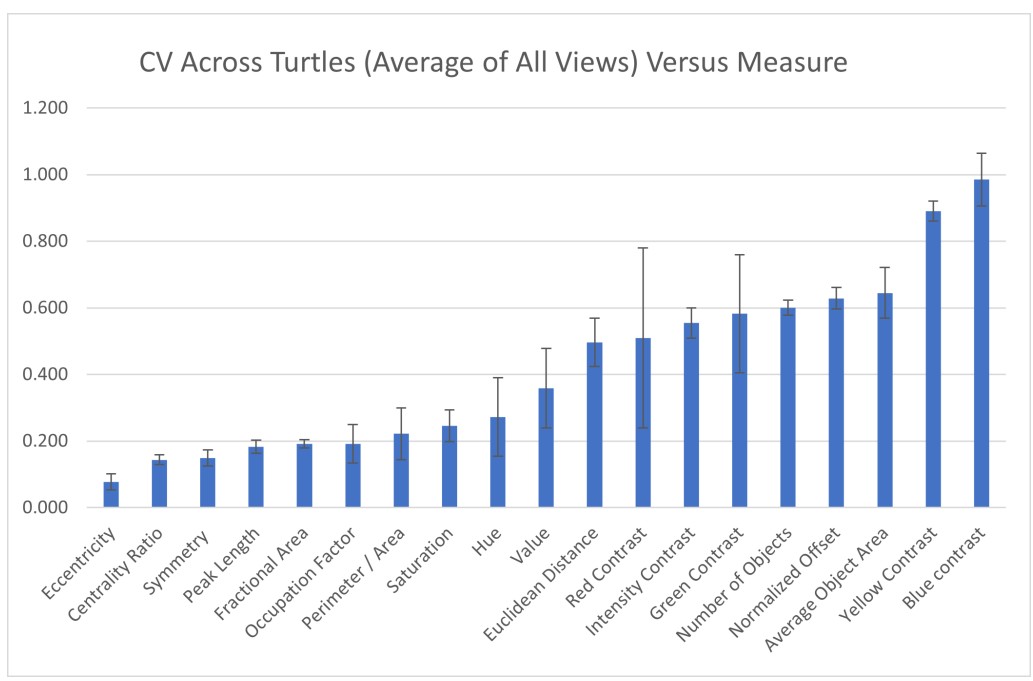

**Figure 7 Variation across individuals.** The coefficient of variation (CV) averaged for all views provides a single CV that captures the variation of the pattern measurement between individual animals. The average CV is shown for all measures ranked from lowest to highest variation. Error bars indicate the standard deviation across views. Full table with all the measurements for different viewpoints can be found in Table S2A.

## Variation in pattern measurements across and within individuals

Across individuals: To assess the measurement variation across all studied individuals, we calculate the mean of the pattern measurements obtained for each individual as the value of the pattern measurement for that turtle and measure the variation of those values across the 55 individuals as $CV = \sigma/\mu$. The coefficient of variation across all the 55 individuals indicate that three measures (blue contrast, yellow contrast, and average area of objects) were more variable across turtles (blue contrast CV = 0.985, yellow contrast CV = 0.890, average area of objects CV = 0.645; Fig. 7). Three measures (mean eccentricity, centrality ratio, and symmetry) were instead the least variable across the 19 measurements (mean eccentricity CV = 0.078, centrality ratio CV = 0.144, and symmetry CV = 0.149; Fig. 7). Full table can be found in Table S2A.

Within individuals: When looking at the CV obtained on the two images taken on the same view of the same individual, six measures had high CVs (higher than 10%): average area and number of objects, yellow, blue, and green contrast, and normalized offset (Fig. 8) while the other measurements had low CVs (CVs < 10%, Fig. 8). This suggests that the majority of the measures are stable when taking repeated images of the same individual without changing the angle or lighting conditions (see below). Additionally, the variation across each measurement for the two photos taken of each individual is similar across the different viewpoints (Table S2B).

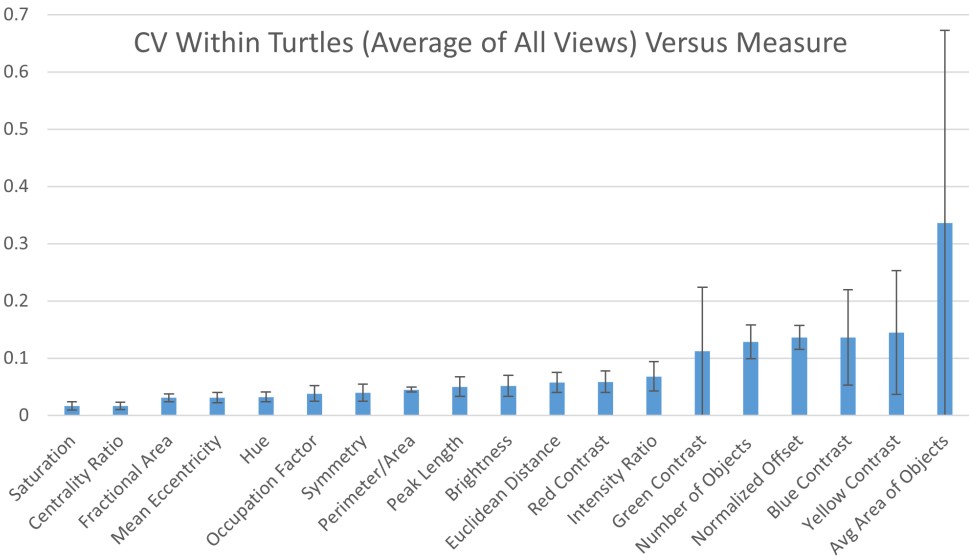

**Figure 8 Variation within individuals.** Pairwise differences were calculated based first on the two images per viewpoint and then normalized by the average across all individuals per viewpoint to yield the coefficient of variation (CV). The CV averaged for all views provides a single CV that captures the variation of the pattern measurement across images within individual animals. The average CV is shown for all measures ranked from lowest to highest variation. Error bars indicate the standard deviation across views. Full table with all the measurements for different viewpoints can be found in Table S2B.

Comparing the within-turtle variability and the across-turtle variability as a ratio is a measure of the extent to which high across-turtle variability may be due to noise (Table S2C). The overall variation attributable to measurement noise was 10 to 30% for most (16 out of 19) measures.

## Variation in pattern measurements due to sample source

Field *versus* museum sampling: To compare how the field sampling *versus* more controlled conditions (Smithsonian specimens) may influence the variation observed for the different pattern measurements, we ran a $T$-test (difference in means), $F$-test (difference in variances), and a second F-test (difference in CVs) across each individual and viewpoint per measurement comparing the field *vs* the museum data. Overall, we found similar *means* across views and pattern measurements independently of where the turtles were sampled ($T$-test, Table S3) though occasionally the top view showed a significant difference ($T$-test with significance $p < 0.05$) (Table 3). Comparing the museum and field samples, the variation of the measures ($F$-test computed on the SD, Table S3) was significantly different for the hue, brightness and red contrast (Table 3). The coefficient of variation of the measures ($F$-test computed on the CV, Table S3) was significantly different for the hue and the occupation factor (Table 3). With the exception of the occupation factor, all measures identified for significant variation for any of the three tests were related to color, especially hue.

**Table 3  Variation due to sample source.** Significant difference was tested between data captured on the Smithsonian ($n = 43$) and field ($n = 12$) collected turtles. A $T$-test (testing means), $F$-test (testing variance) and $F$-test (testing coefficients of variation (CVs)) were conducted and the adjusted $p$-values were recorded. Measures that showed a significant difference ($p < 0.05$) most frequently (for at least three views for any one test) are in the table below. Significant differences of means, sd and cv are indicated with an asterick and a shaded cell. Data for all the measurements and tests can be found in Table S3.

| Pattern measurement | Top view | Right view | Left view | Front view | Back view |
|---|---|---|---|---|---|
| **Hue** | | | | | |
| **Museum Vs. Field $T$-Test, $p$-value** | T(42,11) = −3.17, $p = 0.03$* | T(42,11) = −2.64, $p = 0.15$ | T(42,11) = −2.68, $p = 0.31$ | T(42,11) = −2.61, $p = 0.40$ | T(42,11) = −1.92, $p = 0.50$ |
| **Museum Vs. Field $F$-Test, $p$-value** | T(42,11) = 1.02, $p = 0.94$ | T(42,11) = 10.03, $p < 0.01$* | T(42,11) = 11.37, $p < 0.01$* | T(42,11) = 10.54, $p < 0.01$* | T(42,11) = 9.50, $p < 0.01$* |
| **Museum Vs. Field CV diff., $p$-value** | T(42,11) = 13.76, $p < 0.01$* | T(42,11) = 3.56, $p = 0.09$ | T(42,11) = 1.95, $p = 0.03$* | T(42,11) = 11.12, $p < 0.01$* | T(42,11) = 4.24, $p = 0.20$ |
| **Brightness** | | | | | |
| **Museum Vs. Field $T$-Test, $p$-value** | T(42,11) = −3.46, $p < 0.01$* | T(42,11) = 1.33, $p = 0.44$ | T(42,11) = 1.46, $p = 0.43$ | T(42,11) = 1.07, $p = 0.81$ | T(42,11) = 2.34, $p = 0.40$ |
| **Museum Vs. Field $F$-Test, $p$-value** | T(42,11) = 5.60, $p < 0.01$* | T(42,11) = 3.36, $p < 0.01$* | T(42,11) = 3.49, $p < 0.01$* | T(42,11) = 1.37, $p = 0.69$ | T(42,11) = 2.94, $p = 0.04$* |
| **Museum Vs.. Field CV diff., $p$-value** | T(42,11) = 32.11, $p < 0.01$* | T(42,11) = 2.28, $p = 0.20$ | T(42,11) = 1.66, $p = 0.03$* | T(42,11) = 1.20, $p = 0.80$ | T(42,11) = 13.22, $p = 0.07$ |
| **Red Contrast** | | | | | |
| **Museum Vs. Field $T$-Test, $p$-value** | T(42,11) = −4.17, $p < 0.01$* | T(42,11) = −1.05, $p = 0.52$ | T(42,11) = −0.90, $p = 0.59$ | T(42,11) = −1.27, $p = 0.81$ | T(42,11) = 0.44, $p = 0.74$ |
| **Museum Vs. Field $F$-Test, $p$-value** | T(42,11) = 3.80, $p < 0.01$* | T(42,11) = 3.96, $p < 0.01$* | T(42,11) = 2.84, $p = 0.05$ | T(42,11) = 2.66, $p = 0.10$ | T(42,11) = 3.13, $p = 0.04$* |
| **Museum Vs. Field CV diff., $p$-value** | T(42,11) = 30.23, $p < 0.01$* | T(42,11) = 2.56, $p = 0.94$ | T(42,11) = 1.10, $p < 0.01$* | T(42,11) = 1.14, $p = 0.78$ | T(42,11) = 9.50, $p = 0.99$ |
| **Occupation Factor** | | | | | |
| **Museum Vs. Field $T$-Test, $p$-value** | T(42,11) = 0.02, $p = 0.99$ | T(42,11) = 0.59, $p = 0.63$ | T(42,11) = −0.35, $p = 0.81$ | T(42,11) = −0.06, $p = 1.00$ | T(42,11) = 0.14, $p = 0.94$ |
| **Museum Vs. Field $F$-Test, $p$-value** | T(42,11) = 2.38, $p = 0.22$ | T(42,11) = 1.60, $p = 0.42$ | T(42,11) = 1.26, $p = 0.75$ | T(42,11) = 1.36, $p = 0.61$ | T(42,11) = 1.18, $p = 0.84$ |
| **Museum Vs. Field CV diff., $p$-value** | T(42,11) = 2.95, $p < 0.01$* | T(42,11) = 1.07, $p = 0.22$ | T(42,11) = 4.00, $p < 0.01$* | T(42,11) = 3.71, $p < 0.01$* | T(42,11) = 6.89, $p = 0.99$ |

## Influence of the choice of threshold value

Since the threshold was chosen arbitrarily ("by eye") to capture the pattern, we investigated the effect of changing the threshold by a small and large percentage (Table 4). Increasing the threshold by 5% or 20% decreases the area of the pattern identified, while decreasing (−5%, −20%) the threshold increases the area of the pattern identified (Fig. 6 and Table 4). Figure 6 shows the comparison between the pattern captured by the algorithm at the default 110% threshold (110% more yellow than average in the studied view) to what can be seen by the human eye and in comparison to the increased or decreased thresholds across all five viewpoints. At 110% threshold (our standard in this study), the algorithm captures a large portion of the pattern as seen by the human eye with limited false negative and false positive pattern area. When the threshold is lowered, the fractional area of the pattern is increased since a lower threshold for identifying a pixel as "more yellow than average" is a less strict criterion and leads to the identification of more pattern pixels, and when

**Table 4  Variation of the threshold.** For each view, percent change in the value of fractional area obtained for 110% threshold and thresholds +/−5% and +/−20% as a mean across 55 individuals. Coefficient of variation (CV) for fractional area was used to measure the variation between the optimal 110 threshold and thresholds +/−5% and +/−20%. For each image independently, the baseline threshold of 110% identifies a pixel as a pattern pixel if the pixel is 10% more yellow than the average scute pixel of that image; likewise a threshold of 90% or 130% identifies a pixel as a pattern pixel if the pixel is at least 90% as yellow or 130% as yellow as the average scute pixel.

| Viewpoint | 130% (+20%) | 115% (+5%) | 105% (−5%) | 90% (−20%) |
|---|---|---|---|---|
| **Top view** | %change: −11.7% <br> CV: 0.092 | %change: −3.0% <br> CV: 0.025 | %change: +3.9% <br> CV: 0.029 | %change: +17.9% <br> CV: 0.110 |
| **Left view** | %change: −20.3% <br> CV: 0.170 | %change: −5.3% <br> CV: 0.039 | %change: +6.6% <br> CV: 0.044 | %change: +26.6% <br> CV: 0.158 |
| **Right view** | %change: −19.9% <br> CV: 0.164 | %change: −6.0% <br> CV: 0.046 | %change: +5.5% <br> CV: 0.039 | %change: +24.0% <br> CV: 0.146 |
| **Front view** | %change: −16.2% <br> CV: 0.134 | %change: −4.7% <br> CV: 0.049 | %change: +4.6% <br> CV: 0.043 | %change: +23.7% <br> CV: 0.142 |
| **Back view** | %change: −21.1% <br> CV: 0.181 | %change: −5.5% <br> CV: 0.043 | %change: +6.7% <br> CV: 0.043 | %change: +27.4% <br> CV: 0.142 |

the threshold is increased, the criterion becomes more strict and the fractional area of the pattern is decreased. The impact of the threshold on the value of the fractional area is shown in Table 4. Increasing or decreasing the threshold by 5% of the average value resulted in a modest change in the fractional area of 3–7%. Changing the threshold by 20% in either direction had a larger impact of 12–27% on the fractional area. Variation between the new thresholds and the preferred were found to be fairly consistent across views with the top scutes having less variation. For our criterion that a CV greater than 10% indicates substantial variation, a threshold variation of 5% did not result in substantial variation while a variation of 20% did result in substantial variation. Figure 6 visually presents the changes in thresholds for one of the studied turtles.

## Placement of the color standard

Placement of the color standard at different distances from the object of interest (Fig. S1) resulted in significant variation (CV > 10%) in the brightness measurements. Specifically, adjusting the position of the color standard when taking photos from above and in front led to CVs of 47% and 38%, respectively. This highlights the substantial variability in color measurements that can occur when the color standard is not exposed to the same lighting conditions as the object of interest.

## Influence of the angle at which pictures are taken

Due to the curved shape of the turtle carapace, we tested how small variation in the angle at which the pictures of the animal are taken influences each pattern measurement for each view. We found that out of the seven pattern measurements, the measurements with the highest variation in relation to the CV were the normalized offset, intensity ratio, average area of objects and the Euclidean distance while the three measurements with the lowest variation were the mean eccentricity, occupation factor and centrality ratio (Fig. 9). For added context, the measured coefficient of variation for the average area of objects

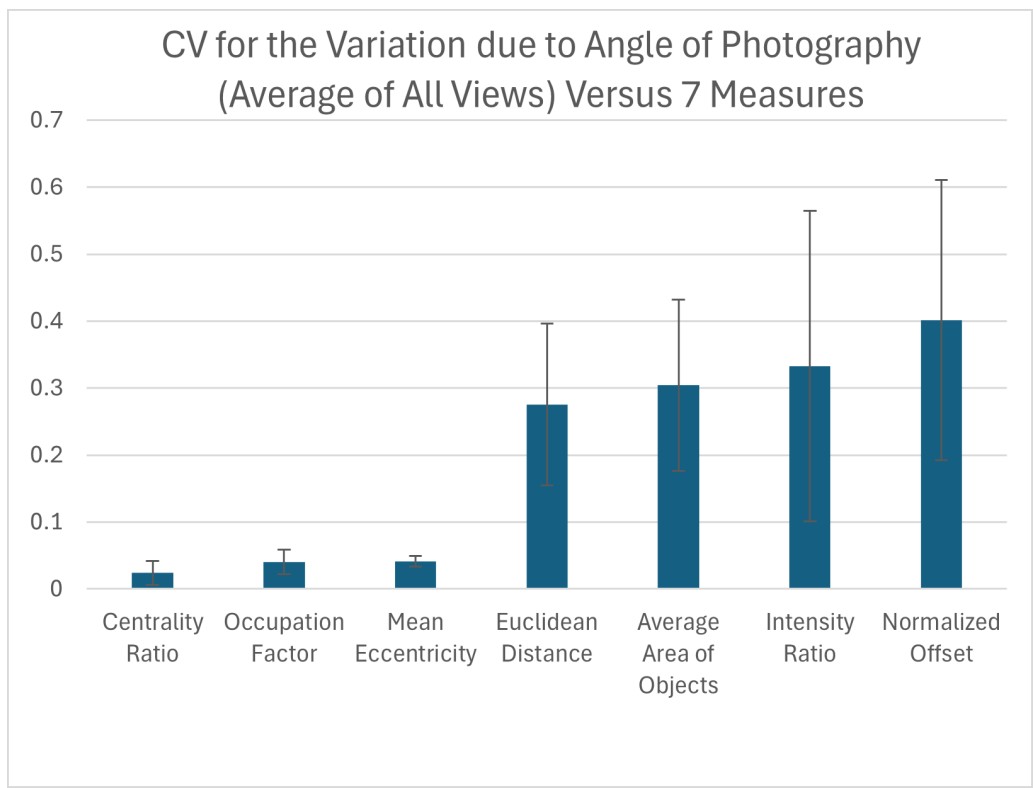

**Figure 9 Variation due to the angle of photography.** The coefficient of variation (CV) averaged for all views provides a single CV that captures the variation of the pattern measurement due to varying the angle of photography. Each viewpoint had three pictures taken per viewpoint per angle: top (−10°, −5°, 0°, 5°, 10°) and front, back, left, and right (0°, 5°, 10°). The average CV is shown for all measures ranked from lowest to highest variation. Error bars indicate the standard deviation across views. The full data set for angle CV for all views for the 7 measures can be found in Table S4.

was comparable to that already measured for within individual turtle variation (Table S4 right-most column), but the normalized offset, Euclidean distance and intensity ratio that were substantially elevated due to angle variation.

When looking at the CV measures for the different viewpoints, there is a clear difference between the top view and the others (front, back, left, and right) (Table S4). For the measures with the highest CVs, the top view generally has the lowest CV with most measurements being under 15%, while for the other viewpoints more than half of the measurements have a CV over 30% (Table S4).

## Categorization and complexity of pattern

We used a consensus score—agreement in categorization of each individual turtle pattern across 31 volunteers—as a measure of pattern complexity, with lower consensus scores suggesting a more complex pattern and vice versa. We found that for the full set of 98 turtles that were categorized, the consensus scores ranged from 66% to 87%, suggesting that the complexity of the pattern varied across turtles since this is a broad range of complexity scores. For the subset of 53 turtles for which we had both consensus scores and
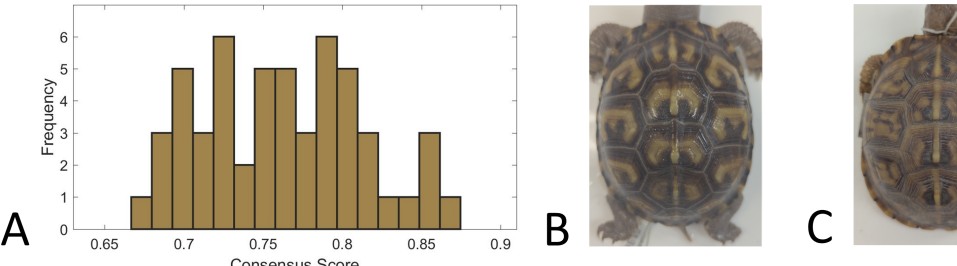

**Figure 10 Consensus scores.** (A) A histogram of the consensus scores for 53 turtles (Smithsonian Natural History Museum, $n = 43$, and field sites, $n = 10$). The mean consensus was 76% (standard deviation of 5%) with 15% ($n = 8$) scoring lower than 70% and 21% ($n = 11$) scoring higher than 80%. (B) Cropped image of the turtle (Smithsonian turtle with ID 519622) with the highest consensus score (87% consensus). 31/31 students labeled the pattern as "M-Shaped" and were in high agreement (greater than 80% consensus) for 8/9 categories. Of the nine categories, the category with lowest consensus was "Banded" (only 68% consensus that the pattern was not "Banded") . (C) Cropped image of the turtle (Smithsonian turtle with ID 139611) with the lowest consensus score (67%). Students were uncertain (less than 60% consensus) on 5/9 categories. The category with highest consensus was "Single-Spotted" (97% consensus that the pattern was not "Single-Spotted").

the measurements used in this work (see 'Citizen science: color pattern categorization to describe complexity'), the consensus scores similarly ranged from 67% to 87% (Fig. 10). Across all the 98 studied individuals, the categories "Patchy" and "Blotchy" had the lowest average consensus scores of 67% and the category "Single-Spotted" had the highest consensus score of 91%. Thus according to this chosen conceptualization of "complexity", patchy and blotchy patterns have the highest complexity and single-spotted patterns have the lowest complexity.

A correlation analysis was performed between the 19 pattern measurements and the consensus score for 53 turtles to determine which measurement(s) may best describe complexity in color pattern in box turtles. Correlation values above 30% were observed for two pattern measurements. The largest correlations were found between the consensus score and the average object area and the eccentricity (Fig. 11). Since we are using higher consensus scores as a measure of lower complexity, this would be interpreted as bigger and less rounded objects being correlated with patterns that are easier to characterize and lower complexity. The largest negative correlation ($-0.28$) was found between the consensus score and occupation factor (Fig. 11). This would be interpreted as a higher occupation factor score being correlated with higher complexity. Although the fractional area was also negatively correlated with consensus ($-0.16$), this correlation was not nearly as strong, suggesting there is an interaction with the contour of the objects since the occupation factor is based on the convex hull of pattern objects. For the 19 measures, none of the correlations were significant after adjusting the $p$-values using the Benjamini–Hochberg correction to control the false positive rate. Cook's distance was run between the measurements and the consensus scores to look for outliers skewing the results. A Cook's distance ($>1$) indicates that one individual influenced the significant results. The Cook's distance was small for these results signifying they are not too influenced by outliers.

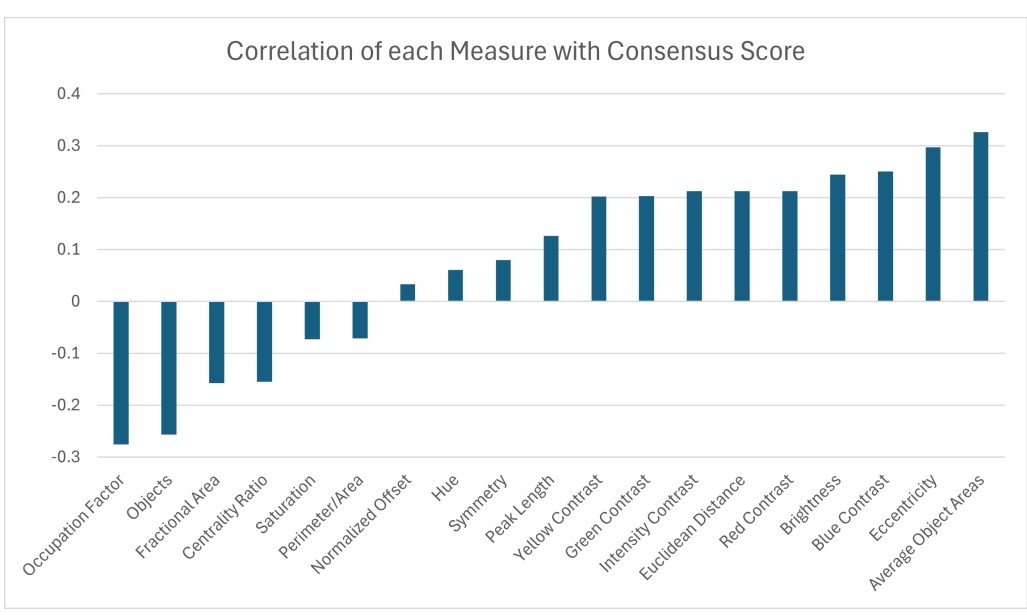

**Figure 11** Correlation coefficients between 19 pattern measurements and consensus scores for 53 turtles (Smithsonian Natural History Museum, $n = 43$, and field sites, $n = 10$).

# DISCUSSION

In this work, we developed a new multi-color threshold-based method to identify and extract complex color patterns. Additionally, we elaborated new algorithms to obtain data on 19 different pattern measurements, capturing as many different aspects of the pattern as possible including coloration and spatial structure of the pattern, object characteristics, and contrast and intensity features of the pattern. The symmetry measure is a novel measure that is particularly useful for animal patterning (*e.g.*, animals often have bilateral symmetry). We applied these methods to the complex color pattern of Eastern box turtles (*Terrapene carolina*) sampled in the field and in a museum collection. We used the data to evaluate how various factors related to capturing and analyzing animal images might affect pattern identification and contribute to variation in pattern measurements. For this reason, images of field turtles were taken of turtles in their natural environment under natural conditions. Finally, we used a pattern categorization approach based on citizen science work to identify which among the 19 pattern measurements used in this study could best depict the complexity of the pattern in box turtles. The framework applied in this study enables the capture of complex animal color patterns and helps identify measurements that best represent this complexity while remaining robust to variations in data collection under non-standardized conditions and during analysis. Despite the importance of studying variation in the organism's color pattern in detail by decomposing it in its measurements, this is still limited due to the general lack of approaches to obtain data on several different measurements of pattern. Advances in this context have been made (*Chan, Stevens & Todd, 2019*; *Glimm et al., 2021*), but they are still largely limited to discrete patterns. As such, our work on the development of an algorithm to extract 19

different pattern measurements capturing the spatial distribution, size and shape, symmetry and color characteristics of the pattern represent an advancement to the field. We found that among the 19 measurements, measures that depend on how the total pattern area is identified as discrete objects (average object area, number of objects) and that depend on the pattern contrast (blue contrast, yellow contrast) are the most variable across the 55 individuals studied in this work (Fig. 7). On the other hand, ratios that result in unitless measures such as fractional area and centrality ratio vary much less (Fig. 7).

To establish how much of this variation could be due to how the images are taken under non-standardized conditions, we used a coefficient of variation calculated across images of the same individuals taken consecutively but independently one from another. Of the three most variable measures, for the blue and yellow contrast measures, only a minor amount of the variation can be attributed to technical aspects of how the images were obtained (Fig. 8). Both contrast measurements had a variability of around 90–100% of which around 15% could be attributed to variation among picture replicates. The high turtle to turtle variability of blue and yellow contrast reflects what can be observed by eye, since one of the most striking differences between the 55 turtle patterns was the range for which the yellow pattern did or did not have a high contrast with the background. Glare in the line of sight due to how the pictures were taken was observed to influence the visibility of the yellow pattern. The curvature of the carapace, as well as the curvature of individual scutes and variation in reflective properties of the shell, likely plays a role in this (see below).

For the two contrast measures with high turtle to turtle variability, the relatively low within turtle variability shows that the measured variability between turtles is an accurate reflection of the pattern variation. Two measures (average area of objects, and mean eccentricity) had an especially high ratio for the relative impact of the measurement noise variation. This shows that the high variability of the average area of objects—the measure with highest turtle to turtle variability other than the contrast measures—is likely largely due to measurement noise variation. Indeed, the average area of objects had a turtle to turtle variability of around 64.5% of which a large fraction (33.6%) could be attributed to variation among picture replicates (Figs. 7 and 8). The high measurement noise in object area underscores the challenge of reliably segmenting patterns with ambiguous boundaries. The large measurement noise in the *average area of objects* (33.6%, Fig. 8) is due to a relatively large measurement noise in the *number of objects* (12.8%, Fig. 8).

Many of the color patterns present in eastern box turtles are non-discrete and complex and pattern objects can often be connected with a region of lower-contrast pixels (Fig. 2C) with difficult to define boundaries. The number of objects of a pattern depends sensitively on whether the algorithm identifies the lower contrast region as pattern pixels connecting what would otherwise be separate objects. Measurements that depend on the number of objects in the denominator (for example, the average area of objects) inherit a high variability. Likewise, the eccentricity is disproportionately affected by this issue since the shapes of objects change depending upon whether they are considered a single or several objects. Although the eccentricity was a measure with especially low turtle to turtle variability (7.8% variability between turtles)—to indicate that this parameter may be relatively conserved in this species—it had a relatively high ratio (40%) of measurement

variability (noise) relative to the turtle variability (Table S2), suggesting that the observed variation is largely due to noise in the measurements. Measures that are less affected by the challenge of defining object boundaries within a pattern are those that do not rely on dividing the entire patterned region into discrete objects. Examples include total fractional area and the ratio of perimeter to area.

Additionally, the carapace curvature adds noise to measurement variation, as the top scutes show less variability on average than other scutes when the photography angle and threshold are adjusted (Table 4 and Fig. 9). Top scutes are relatively flatter than the other four viewpoints used (Fig. 5). Among the 19 measurements used in this work, we developed an algorithm to extract information on how symmetric the scute pattern within each view is. The symmetry of the pattern has long been identified as an important factor for both natural and sexual selection and has often been tested in predation studies (*Cuthill, Hiby & Lloyd, 2006*; *Enquist & Arak, 1994*; *Forsman & Herrström, 2004*; *Forsman & Merilaita, 1999*). However, symmetry has been challenging to quantify and measure. In fact, studies are often theoretically estimating the impact of symmetry without using empirical data and measures of symmetries of the pattern obtained on organisms (*Cuthill, Hiby & Lloyd, 2006*; *Forsman & Herrström, 2004*). However, some studies have also used fractal geometry to describe self-similarity and symmetry in butterfly wings (*e.g.*, *Otaki, 2021*). The algorithm developed for this study to infer the degree of symmetry of the color pattern is based on the maximum achievable area of overlap between the original pattern and its mirror image under rotations and translations. We found that the range (min–max) of the mean symmetry for the five views was (0.54–0.75) with a CV measure of 15% (Table S2A). This suggests that the symmetry of all scutes is on average over 50%, but below 75%, meaning the pattern generally has a moderate level of symmetry across the carapace and that overall is not highly influenced by noise due to how the images are obtained.

Our results also highlight the importance of obtaining brightly colored images to study color patterns in order to minimize variation due to how the measurements are taken. When data from turtles sampled in the field were compared to turtles samples at the Smithsonian, we found consistently for many of the color measures, significant differences among the SD and CV ($F$-tests and CV diff, $p$-values < 0.05, Table 3 and Table S3) even though the means were not significantly different. A likely cause of this is that images obtained in the field were on living individuals with more vibrant patterns than the museum specimens. More vibrant colors of living specimens enhance pattern-background contrast and improve the algorithm's reliability in detecting pattern objects.

The pattern algorithm developed for this study utilized a threshold to distinguish a pattern that exhibited a yellower hue than the average of all pixels for the studied region (the scute). The threshold, a critical component of the algorithm, was employed to isolate pixels that deviated by a specific percentage from the average yellow value. By utilizing fractional area, a highly reliable pattern measurement, we assessed the algorithm's sensitivity (Table 4) to determine the extent of variation in pattern identification when the threshold was adjusted by 5% or 20%. We found that a 5% change to the threshold of the pattern capture algorithm for image binarization had little impact on the value of fractional area, while twenty percent change had a clear impact on the pattern recognized. We found that

the coefficient of variation for the fractional area also increased, suggesting that the chosen threshold for capturing the pattern was also optimized for minimizing variation, capturing well the "true" pattern rather than noise artifacts.

Our multi-color pattern extraction method is fully automated; however, its development was labor-intensive, and our findings indicate that pattern details of the automated extracted pattern are sensitive to environmental factors. The fact that the process is fully automated allows application to much larger datasets in the future. While fully functional, aspects of the process could be made more efficient, such as developing a more sophisticated algorithm for the symmetry index instead of an effective but slow brute force search through all possible translations and rotations of the pattern. Certain distortion effects due to the curvature of the scutes can be addressed by reconstructing three dimensional representations instead of two dimensional images. We are also extending this work by comparing its performance with deep learning approaches, trained on both automated and hand-labeled datasets. Such an investigation will assess whether deep learning can reduce the laborious aspects of the extraction process while accurately capturing pattern features,and showcase the potential of emerging deep learning techniques in this domain.

Finally, we used a citizen science approach to understand how people visually estimate the level of complexity in the color pattern of the studied box turtles. The level of complexity was inferred from the overall consensus score based on 31 independent volunteers and obtained across the provided classification categories of color pattern. Lower consensus scores indicated that the pattern was more complex and more difficult to categorize, while higher consensus scores signified more easily identifiable, potentially simpler, patterns. We note that complexity of pattern can be defined in many different ways, and lack of agreement in description is one measure among many potential metrics. For instance, a different measure of complexity, 'algorithmic complexity' (*Kolmogorov, 1965*), has been assessed by examining the file size of a compressed graphic interchange format (GIF) that stores a pattern, a method previously used as a measure of pattern complexity (*Chan, Stevens & Todd, 2019*). Our choice to use the consensus-of-description method was driven by its straightforward implementation (each volunteer simply assigned a binary value to each pattern description) and the ability to capture a wide range of perceptions since the pattern descriptions were chosen interactively and were not mutually exclusive, thus organically reflecting the diverse opinions of the participants. In contrast, asking volunteers to assess the complexity of a pattern directly would require each volunteer determining independently what is meant by complexity of pattern, with potentially inconsistent results, or would require the imposition of a predefined, "top-down" definition of complexity.

Across the categorized 98 turtles, consensus scores varied from 66% to 87% depending on the individual animal pattern. A consensus of 67% is a relatively low score, considering that a consensus of 50% would be the score of maximally divided students, and a score of 67% means that on average only two out of three students agreed on the appropriateness of each category. Thirteen turtles (13%) with scores lower than 70% support the complexity and the difficulty of categorizing the color pattern of the Eastern box turtles. In contrast, 15% of turtles with high consensus (>80% average consensus for all categories) reflect agreement

on most categories for these turtles. Patterns with the highest consensus scores were "M-shaped" or "single-spotted", while "banded" patterns had lower consensus scores. We then determined which pattern measurements best capture the complexity of the color patterns. Measurements that were found to positively correlate with the complexity (that is, with a decrease in consensus) was the occupation factor, while measurements that displayed negative correlations to complexity were the average object area and eccentricity. The interpretation of these correlations requires some care. For example, it is not necessarily the case that these measures intrinsically correlate (or anti-correlate) with complexity. Rather it is likely that certain turtle pattern phenotypes correlate with complexity, and that these patterns have distinctive measurements that influence the correlations. For example, if "M-shaped" patterns typically have high consensus, then the distinctive measurements properties of M-shaped patterns, whatever they may be, will anti-correlate with complexity.

We do not study pattern variation among *Terrapene carolina* turtles here, but develop methods that can be applied to studying that variation using digital photographs. We provide a summary of several "best practices" in Table S5. The methods produced in this research are not only relevant to study color pattern variation in *Terrapene carolina*. While this algorithm was developed to identify a distinct pattern seen on this species, the pattern recognition algorithm can be modified relatively easily to identify other color patterns that have distinct colorations and a relatively high contrast to the background coloration. As such the identification and extraction algorithm, as well as the codes to obtain measures on the pattern measurements can be highly customizable and applied to study color pattern variation in other organisms. Additionally, this work provides a pipeline to infer the influence of variation in how images are obtained and data are extracted on variation in the algorithm and the studied pattern measurements, something that should be incorporated also in future studies on color pattern variation. Although our study focused on *Terrapene carolina*—a species benefiting from the stability of a hard shell—the algorithm is inherently adaptable to other taxa with distinct color patterns, such as those available from open-access platforms like iNaturalist. Extending the methodology for other contexts and especially to species with flexible bodies, would require extensive data cleaning and modifications to account for distortions introduced by body movement and variable imaging conditions.

## ACKNOWLEDGEMENTS

We are thankful to Andrew Eberly and Bert Harris for access to carrying out sampling at the Clifton Institute and for facilitating training in data collection on box turtles. We are grateful to Addison Wynn, Esther Langan, and Christina Keating Sami at the Smithsonian Institution for granting access to the box turtle collection. We are thankful to Jessica Meck, Max Earle, Jordan Davis, Kathryn Long, Paige Saari, Naomi Whiteside, Riley Moreau, and Marcela Alvarenga for their help with the field work. Finally, we are thankful to Daniel Hanley and Patrick Gillevet for providing valuable comments on an early version of this manuscript. We would like to acknowledge the use of AI. After prompts by the authors, ChatGPT (OpenAI, 2024) refined the earliest draft of the code for the symmetry measure in MATLAB (version R2020a); provided snippets of Excel script that were used to compile

data from multiple worksheets; provided the initial draft of the black cartoon used in Fig. 2A, was used for coding assistance and debugging for the software environment R (*R Core Team, 2024*). All edits were critically reviewed by at least one of the authors. We emphasize that the AI did not contribute to the generation of research ideas, data analysis, or the interpretation of results. All intellectual contributions and critical interpretations remain the responsibility of the authors.

### Funding

Christopher Wicker and Amanda Daisey provided funding for fieldwork. Data collection expenses were supported through funding from George Mason University to Ylenia Chiari. The funders had no role in study design, data collection and analysis, decision to publish, or preparation of the manuscript.

### Grant Disclosures

The following grant information was disclosed by the authors:
George Mason University.

### Competing Interests

The authors declare there are no competing interests.

### Author Contributions

- Erik Maki performed the experiments, analyzed the data, prepared figures and/or tables, authored or reviewed drafts of the article, and approved the final draft.
- Tilmann Glimm conceived and designed the experiments, performed the experiments, analyzed the data, authored or reviewed drafts of the article, and approved the final draft.
- Paramahansa Pramanik analyzed the data, authored or reviewed drafts of the article, and approved the final draft.
- Ylenia Chiari conceived and designed the experiments, authored or reviewed drafts of the article, organized and coordinated data collection in the field and for the museum, provided funding for data collection, and approved the final draft.
- Maria Kiskowski conceived and designed the experiments, performed the experiments, analyzed the data, prepared figures and/or tables, authored or reviewed drafts of the article, and approved the final draft.

### Animal Ethics

The following information was supplied relating to ethical approvals (i.e., approving body and any reference numbers):

George Mason University IACUC committee (permit number 1908275).

### Field Study Permissions

The following information was supplied relating to field study approvals (i.e., approving body and any reference numbers):

Permission was granted to work at the Clifton Institute without the requirement of a permit. United States Fish and Wildlife provided a permit to work at Mason Neck Wildlife Refuge (permit number 51600-22RES05).

## Data Availability

Raw study data and analyses for figures and tables are available in the Supplementary Materials.

The digital photos of turtles (original dng files for 5 views per turtle, png files with scutes outlined) are available at Zenodo:

Maki, E., & Kiskowski Byrne, M. A. (2025). Digital Images of Eastern Box Turtle (Terrapene carolina) from Clifton Institute, Mason Neck Wildlife Refuge and Smithsonian (Data set). Zenodo. https://doi.org/10.5281/zenodo.15373795.

## Supplemental Information

Supplemental information for this article can be found online at http://dx.doi.org/10.7717/peerj.19690#supplemental-information.

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
