# Peer review of "New approaches for capturing and estimating variation in complex animal color patterns from digital photographs: application to the Eastern Box Turtle (Terrapene carolina)"

_PeerJ, doi:10.7717/peerj.19690_

## Round 0.1 · original submission · Major Revisions

Please, take all the reviewers' suggestions into account and address their concerns.

Reviewer 1 ·

Basic reporting

No comment

Experimental design

No comment

Validity of the findings

No comment

Additional comments

The manuscript described a new approach for estimating variation in complex color pattern in turtles. The authors collected images of different turtles on field and museum under different conditions: lighting, angle, camera position, etc. The application of series of digital image processing techniques to estimate the complex color pattern is appreciable. However, the manuscript lags in certain justifications and explanation, and needs to be improved and revised majorly by addressing the following comments:
1. It is always better to align the Tables and Figures along with the text for better and quick readability.
2. The necessity of using Table 1 is unclear. That is, if there is only one row, it is suggested to explain in the text rather a Table.
3. Figure 1 shows the example of species with complex color patterns however, the difference from simple pattern is not provided. Hence, it would be better to differentiate the species with simple and discrete pattern with the complex pattern by displaying some images.
4. In Figure 2, green circle mentioned in the text is not visible. Is it confused with a red circle? Confirm. Also, Figure 2, Panel B is not clear. Provide a better clarity picture.
5. The camera settings used for data collection/image acquisition mentioned in line 143 can be tabulated for better readability. That is, avoid mentioning the values in the text instead tabulate them.
6. Have you referred to any literature to collect photos from 5 views? If yes, please cite the article referred.
7. Figure S1 used as a supplementary material can be used in the manuscript.
8. What is CV mentioned in Figure S3 caption? Is it a standard abbreviation?
9. The sentence in line 194 is unclear.
10. What is GIMP, HSB in lines 193 and 195, respectively? Their full-forms are missing. It is suggested to give the full-form of an abbreviation once in the text and later, use the corresponding abbreviation. Also, listing all the abbreviations used in the manuscript using a Table is a good practice.
11. A pseudocode can better interpret an algorithm or methodology. Hence, the pseudocode of the color pattern identification algorithm is strongly recommended.
12. The step-by-step resultant images explained in Section 2.2.4: color pattern identification algorithm is recommended.
13. The description of pattern measurements given in Table 2 is appreciable. However, it is suggested to align the Table columns to either right, left, center, or justify. Also, is it required to measure the red, blue and green contrast? Because, it was mentioned that the region of interest or the data collected are with respect to the yellow patterns. Also, the algorithms are correcting the color contrast. Therefore, the measurement of the yellow contrast would be sufficient and the other measures are unnecessary. Justify.
14. In line 290, the right hand side and left hand side of an equation are missing. It is suggested to present an equation in the correct form with an equation number. (Use the equation number to in the text whenever necessary.)
15. Initially, it is mentioned as 55 individual turtles. But the correlation test is run on the 53 individuals. The authors are recommended to clarify the confusion. Also, the justify the test results with the correlation values.
16. In line 381, why again confusing the readers with the number 550? Instead directly specify that a total of 610 pictures are analyzed. Justify why 550 when it is 610? Is it a typo? If yes, the authors are strongly recommended to read the manuscript before submitting to avoid such confusions.
17. Check the equation given in line 395. Correct the mistake.
18. Again in line 510, it is mentioned as 98 turtles. Totally, how many field and museum turtles used in the data collection and how many of them are used in the analysis?
19. In the author contribution, it is suggested to give the author names followed by their respective contributions.
20. The manuscript highlights the method to estimate the complex color pattern in turtles. However, the purpose behind it is not provided. I suggest the authors should explain the application/need/purpose of estimating color pattern in turtles. Explain how your study is useful to the society, in other words, mention its social impact.
21. Neither the study on animal color pattern nor the pattern measurement techniques used in this manuscript are novel. However, the efforts in data collection and the application od DIP techniques on the turtle images are appreciable. Although the authors are recommended to list the major contributions of your work.
22. The authors should list the limitations of the present research work and how you will try to address them in future.
23. The methodology seems to be very tedious and complex. Why didn’t you use the deep learning methods when you have the facility of data collection? Unlike the methodology used in the current manuscript, deep learning can provide more quick and robust results as they are not bounded/restricted to certain threshold values.
24. It is always better to test your methodology on any other openly available datasets to evaluate the generalization ability of the proposed method. Therefore, justify the lines 685-693 by evaluating the proposed methodology with other open access datasets.
25. The authors are also recommended to check for grammatical errors.

Reviewer 2 ·

Basic reporting

The paper has value and contains useful information for those in the field of photography and optics, and perhaps, even more widely. However, I have concerns about the way information is displayed in figures and tables, and also how the text describes aspects of the study. Below are my comments for the authors to consider:

I think just the use the species name Terrapene carolina is sufficient and not the subspecies carolina as the paper includes the geographic location for the wild animals. Where were the Smithsonian animals collected from? Also, (if not already done) please include museum ID #s for all of those photos so future researchers can match them to the exact specimens.

I suggest the title should be something more like "Factors influencing variation in complex color patterns estimated from photographs: A case study with the Eastern Box Turtle (Terrapene carolina)"

Sources for sex determination: Ernst et al., 2008 = this is for Asian box turtles, not North American species; Weiss, 2009 = This is not a reliable source. I am skeptical of the use of these sources, especially when there are many good ones out there for this species.

Why were animals measured with calipers? I do not see these data anywhere.

Figure 1: I think that B, C, and D should be replaced with images of other species of terrestrial turtles, even other North American Box Turtle species, such as T. ornata, T. major, and T. triunguis. The reasons for the use of the butterfly and fish are not clear. Also, the quality of these images varies markedly with some clear and others blurry, plus the zoom level is different, which makes this figure not useful in its current state.

Figure 2: The image in B is very blurry and of low quality. This entire paper is based on images yet uses poor images in the first four figures. And B is a red circle, not green. This figure needs to be improved a lot.

Figure 3: Were these color or light balanced? They appear to be very different in whiteness and shading, which makes me wonder about the analyses. Also, the turtle shells are not all clean, so some of these may look different based on dirt or debris. This figure should be used to define the different patterns that you used later in the paper for people to identify. Just showing the variation is not very useful without categories or more discussion about the image differences for their impact on the analyses. Examples of different parameters (angles, etc.) would be more useful here.

Figure 4: This image is also of low quality. Were these really the images that were used in this study? I am confused because the study states that photographing conditions were very controlled for and the entire study is based on the quality of the photos of turtles, but all of the images of turtles in the figures seem to be poor quality.

Figure 5: This is not rotated vertically and the circle encompasses more than just one scute, so I am unsure what value this figure is adding to the study. Was just 1 scute identified per animal for analyses and were these the same scute for each animal? If so, the figures 1-5 are not clear.

Figure 6: This is the first figure that I think is valuable and actually helps the readers understand what was done is this study. I do not understand the first 5 figures and the images are of very low quality or not clear why they are included. Also, why are some of the green lines identifying scutes cut off in the images? That should be fixed.

Figure 7: The second useful figure. However, B and C look very dark, and B is shiny (presumable with unevaporated ethanol) but C is not. Was this something controlled for?

Figures S1, S2: Put these into the main manuscript in replace of Figures 1-5 as they are much more informative of what was actually done in this study and are of much better quality.

Figure S3: Consider adding this to the main manuscript too.

Table 1: This has just one row of information, so does not warrant a table. Just include this as a definition in the paper.

Table 2: Very useful. I would like to see a figure like Figure 6 that shows how these variables vary across more images, especially the ones that had the biggest impact on the algorithms.

Table 3: Very useful. I would like to see a figure like Figure 6 that shows how these variables look in the turtles, especially the ones that were most identified by students.

Table 4: These would be better visualized as a histograms with error bars.

Table 5: Reduce to include only significant differences, and this would be better visualized as histograms with error bars, showing the ones that were significant.

Tables 6-8: Keep 6 and 8. Consider displaying 7 as histogram with error bars and combine with Table 4.

Experimental design

Did you use adjusted p-values for multiple testing with 19 correlations per animal?

Are you sure that just 10 animals from the field is indicative of the pattern variation within that population?

What was the geographic source location for the 43 turtles at the Smithsonian?

What about sex-based color pattern differences? Did you account for sex in your analyses? Did you note the sex of turtles in the field or museum?

The turtle in Figure 7 looks small and is perhaps a juvenile--what about ontological differences in color pattern? It states you measured turtles but those data are not presented anywhere I could find.

Why is the pipeline in MATLAB but analyses were done in R? More info is needed.

Validity of the findings

The findings appear robust. However, they are overexplained and include many methodological-like statements that belong in the methods section. For example, in Results section 3.5, the first three sentences explain methods but no results. There are many more examples throughout the Results like this. Results should focus on the findings and not repeat information listed elsewhere. A similar problem (but not as severe) exists in the Discussion where there are a lot of Results duplicated here that are already listed in that section.

My biggest concern is the figures, which in their current form are inadequate and do not add value. I have made a number of suggestions in the above sections that would significantly improve the manuscript if adopted.

The conclusions need to focus more on turtles and specifically North American Box Turtles as this study is most useful for them. I would like to see more about how this could be used to compare pattern variation of populations across the species range and within populations at specific sites.

Readers would benefit from a more clear step-by-step best guidelines for how to implement this method to study color pattern in turtles. Based on all these results, I would like to see a easy-to-understand summary Table that includes the top recommendations on what to do, what equipment to use, and best steps to generate the images and analyze the color data within them.

Additional comments

I think that the text is too long in some sections because information is repeated in different sections and also included in very long tables. The authors should consider ways to display their tabulated results in a more concise, and perhaps graphical manner, which would increase the impact of this manuscript.

---

## Round 0.2 · accepted · Accept

Thank you for addressing all reviewers' comments. I am happy with the current version and recommend it for publication.

Reviewer 1 ·

Basic reporting

The authors have addressed the review comments and suggestions and incorporated them into the manuscript. Especially, the figure qualities are improved, the step-by-step procedure has made the understanding of the algorithm easier, the insight of the present work is well-described, and the limitations and the scope for future work are justified. The updated manuscript is recommended for publication. However, the following comments need to be taken into consideration:
1. Check for grammar and maintain the tense throughout the manuscript.
2. Generally, an equation number is assigned to an equation irrespective of one or more equations. I strongly recommend giving the equation number.
3. The contributions of the manuscript are scattered. I strongly recommend listing them to leverage the understanding and readability.

Experimental design

No comment

Validity of the findings

No comment

Additional comments

-